# Towards Bridging Generalization and Expressivity of Graph Neural Networks

**Shouheng Li**[1,4]**, Floris Geerts**[2]**, Dongwoo Kim**[3]**, Qing Wang**[1]

[1] School of Computing, Australian National University, Australia

[2] Department of Computer Science, University of Antwerp, Belgium

[3] CSE & GSAI, POSTECH, South Korea

[4] Data61, CSIRO, Australia

shouheng.li@anu.edu.au, floris.geerts@uantwerp.be
dongwoo.kim@postech.ac.kr, qing.wang@anu.edu.au

## Abstract

Expressivity and generalization are two critical aspects of graph neural networks (GNNs). While significant progress has been made in studying the expressivity of GNNs, much less is known about their generalization capabilities, particularly when dealing with the inherent complexity of graph-structured data. In this work, we address the intricate relationship between expressivity and generalization in GNNs. Theoretical studies conjecture a trade-off between the two: highly expressive models risk overfitting, while those focused on generalization may sacrifice expressivity. However, empirical evidence often contradicts this assumption, with expressive GNNs frequently demonstrating strong generalization. We explore this contradiction by introducing a novel framework that connects GNN generalization to the variance in graph structures they can capture. This leads us to propose a $k$-variance margin-based generalization bound that characterizes the structural properties of graph embeddings in terms of their upper-bounded expressive power. Our analysis does not rely on specific GNN architectures, making it broadly applicable across GNN models. We further uncover a trade-off between intra-class concentration and inter-class separation, both of which are crucial for effective generalization. Through case studies and experiments on real-world datasets, we demonstrate that our theoretical findings align with empirical results, offering a deeper understanding of how expressivity can enhance GNN generalization.

## 1 Introduction

Graph Neural Networks (GNNs) (Scarselli et al., 2008) have become pivotal in modern machine learning, anchored in two main pillars: *expressivity* and *generalization*. Expressivity refers to a GNN's capacity to distinguish between diverse graph structures, thereby determining the scope of problems it can address (Xu et al., 2019; Morris et al., 2019). Highly expressive GNNs can capture intricate dependencies, essential for tasks like molecular property prediction (Gilmer et al., 2017), drug discovery (Gaudelet et al., 2021), and protein-protein interaction prediction (Zitnik et al., 2018), where minor structural variations have significant implications. Generalization, on the other hand, reflects a GNN's ability to transfer learned knowledge to unseen graphs. Given the diversity in graph structures, sizes, and complexities, GNNs that generalize well maintain consistent performance across varying datasets. Together, these properties enable GNNs to model complex graph structures while remaining effective across new, unseen data, making them invaluable for graph-based analysis.

Theoretically, a trade-off is expected between expressivity and generalization: highly expressive models can capture complex graph structures but may overfit and generalize poorly without proper regularization. Conversely, models focused on generalization often sacrifice some expressivity to perform better across diverse, unseen graph structures. Recent work indeed shows a strong correlation between a GNN's VC dimension and its ability to distinguish non-isomorphic graphs (Morris et al., 2023). *A more nuanced theoretical analysis is needed*, however. Indeed, *empirical evidence frequently contradicts the above view.* Highly expressive models often exhibit strong generalization performance

in practice (Bouritsas et al., 2023; Wang et al., 2023). In the restricted context of linear separability, margin-based bounds offer partial alignment between theory and practice (Franks et al., 2024), yet our broader understanding of how expressivity influences generalization remains incomplete. This raises two key questions: (i) *How does the structured nature of graphs affect GNN generalization?* (ii) *How does a GNN's expressivity influence its ability to generalize across tasks and unseen data?* Addressing these questions is vital for advancing GNN applications in real-world scenarios.

**Present work.** Building on the foundational work of Chuang et al. (2021), we explore how the *concentration* and *separation* of *learned features*, key factors in multiclass classification generalization, translate to graph-based models. Their bound, derived from $k$-variance (Solomon et al., 2022) and the expected optimal transport cost between two random subsets of the training distribution, motivates our adaptation to graph embeddings. Leveraging these insights, we extend their framework to capture the structural properties of graph embedding distributions and contribute the following:

- For arbitrary graph encoders, including GNNs, we show that their generalization can be bounded in terms of the generalization bound of any more expressive graph encoder. This allows capturing structural properties of graph embedding distributions with respect to their bounding encoders.

- Under certain margin conditions, we demonstrate that the downstream classifier generalizes well if (1) embeddings within a class are well-clustered and (2) classes are separable in the embedding space in the Wasserstein sense, extending Chuang et al. (2021)'s results to graphs.

- On the real-world PROTEINS dataset (Morris et al., 2020a), we empirically show how a more expressive model influences generalization by measuring variance in graph embedding distributions.

- We apply the empirical sample-based bound of Chuang et al. (2021) to graph classification tasks, verifying that empirical findings align with our theoretical insights, thus demonstrating the applicability of our approach to predict generalization.

Our results offer a flexible framework for analyzing generalization properties of complex graph encoders via simpler encoders, such as those based on 1-WL, its higher-order variants $k$-WL (Cai et al., 1992; Grohe, 2017), homomorphism counts (Zhang et al., 2024), or $\mathcal{F}$-WL (Barceló et al., 2021), provided they upper-bound the encoders under consideration. *Overall, we present a versatile tool for evaluating whether increased expressiveness improves or worsens generalization.*

## 2 RELATED WORK

Regarding generalisation of GNNs, Scarselli et al. (2018) utilize VC dimension to study the generalization of an older GNN architecture, distinct from modern MPNNs (Gilmer et al., 2017). Garg et al. (2020) show that the Rademacher complexity of simple GNNs depends on maximum degree, layer count, and parameter norms, while Liao et al. (2021) develop PAC-Bayesian bounds relying on node degree and spectral norms; see Karczewski et al. (2024) for extensions. Improved bounds using the largest singular value of the diffusion matrix are proposed by Ju et al. (2023). Transductive PAC-Bayesian bounds for knowledge graphs are discussed by Lee et al. (2024). Random graph models are leveraged by Maskey et al. (2022), who show GNN generalization improves with larger graphs. Connections between VC-dimension and the 1-WL algorithm are made by Morris et al. (2023), who bound it by the number of 1-WL colors. Levie (2023) provide bounds based on covering numbers and specialized graph metrics.

For GCNs, Verma and Zhang (2019) derive generalization bounds using algorithmic stability, with Zhang et al. (2020) focusing on single-layer GCNs and accelerated gradient descent. Zhou and Wang (2021) extend this to multi-layer GCNs, showing that generalization gaps increase with depth. Similarly, Cong et al. (2021) highlight this trend in deeper GNNs and propose detaching weight matrices to improve generalization. Further analyses of transductive Rademacher complexity using stochastic block models are offered by Oono and Suzuki (2020); Esser et al. (2021). Tang and Liu (2023) establish bounds involving node degree, training iterations, and Lipschitz constants, while Li et al. (2022) study topology sampling and its impact on generalization. Lastly, Franks et al. (2024) explore margin-based bounds.

Moving to the expressivity of GNNs, MPNNs' expressivity is bounded by 1-WL (Xu et al., 2019; Morris et al., 2019), showing the need for more expressive methods. Many such models have been put forward. For example, the $\mathcal{F}$-MPNNs (Barceló et al., 2021) enhance expressivity via homomorphism counts, similar to Bouritsas et al. (2023). Homomorphism counts have become a popular mechanism in graph learning (Nguyen and Maehara, 2020; Welke et al., 2023; Zhang et al., 2024; Jin et al., 2024; Lanzinger and Barcelo, 2024), and will be central to our analysis. Additional discussion on related work can be found in Appendix A.

## 3 PRELIMINARIES

**Graphs and homomorphisms.** We begin by considering undirected graphs $G = (V_G, E_G)$, where $V_G$ represents the set of *vertices* and $E_G \subseteq V_G \times V_G$ forms the *edge* set, a symmetric relation. For any vertex $v \in V_G$, its set of *neighbors* is given by $N_G(v) := \{u \in V_G \mid (v, u) \in E_G\}$. A *homomorphism* from a graph $G$ to another graph $H$ is a mapping $h : V_G \to V_H$ such that each edge $(v, w) \in E_G$ is mapped to an edge $(h(v), h(w)) \in E_H$. An *isomorphism*, on the other hand, is a bijective function $f : V_G \to V_H$ that preserves adjacency: $(v, w) \in E_G$ if and only if $(f(v), f(w)) \in E_H$. The notation $\mathsf{Hom}(G, H)$ refers to the *number of homomorphisms* from $G$ to $H$, and the function $\mathsf{Hom}_G(\cdot)$ maps any graph $H$ to $\mathsf{Hom}(G, H)$. Given a sequence $\mathcal{F} = (F_1, F_2, \ldots)$ of graphs, we define $\mathsf{Hom}_{\mathcal{F}}(\cdot)$ as $(\mathsf{Hom}_{F_1}(\cdot), \mathsf{Hom}_{F_2}(\cdot), \ldots)$, a tuple of homomorphism counts. A *graph invariant* is any function $\xi$ on graphs that is unchanged under isomorphisms, i.e., $\xi(G) = \xi(H)$ when $G$ and $H$ are isomorphic. For instance, $\mathsf{Hom}_{\mathcal{F}}(\cdot)$ serves as a graph invariant for any graph sequence $\mathcal{F}$. Moreover, we introduce the concept of *rooted graphs*, where each graph $G^r$ has a distinguished root vertex $r \in V_G$. For two rooted graphs $G^r$ and $H^s$, a homomorphism must also map the root $r$ of $G$ to the root $s$ of $H$. The notation $\mathsf{Hom}_{F^r}(\cdot)$ captures the number of homomorphisms $\mathsf{Hom}(F^r, G^v)$ from a rooted graph $F^r$ to any rooted pair $(G, v)$, where $v$ is treated as the root of $G$. Similarly, $\mathsf{Hom}_{\mathcal{F}^r}(\cdot)$ is defined, capturing important *vertex invariants* for pairs $(G, v)$. We illustrate some of the above concepts by examples in Appendix B.

**Graph neural networks and WL.** We extend the above notions to *featured graphs* $G = (V_G, E_G, \zeta_G)$, where each vertex is endowed with a feature vector $\zeta_G : V_G \to \mathbb{R}^{d_0}$ of some fixed dimension $d_0 \in \mathbb{N}$. We focus on Message-Passing Neural Networks (MPNNs) (Gilmer et al., 2017), enhanced with homomorphism counts from $\mathcal{F}$ (Barceló et al., 2021). For a sequence of rooted graphs $\mathcal{F} = (F_1^r, F_2^r, \ldots)$, the initial vertex representation for a vertex $v \in V_G$ in an $\mathcal{F}$-MPNN is:[1]

$$\phi_{\mathcal{F}}^{(0)}(G, v) := (\zeta_G(v), \mathsf{Hom}(F_1^r, G^v), \mathsf{Hom}(F_2^r, G^v), \ldots).$$

At each iteration (*layer*) $0 \le \ell \le L$, this representation is updated as follows:

$$\phi_{\mathcal{F}}^{(\ell+1)}(G, v) := \mathsf{upd}^{(\ell)}\left(\phi_{\mathcal{F}}^{(\ell)}(G, v), \mathsf{agg}^{(\ell)}\left(\{\!\{\phi_{\mathcal{F}}^{(\ell)}(G, u) \mid u \in N_G(v)\}\!\}\right)\right),$$

where $\{\!\{\cdot\}\!\}$ denotes a multiset, and $\mathsf{upd}^{(\ell)}$ and $\mathsf{agg}^{(\ell)}$ are differentiable *update* and *aggregation* functions, respectively. After $L$ iterations, a final pooling operation produces the graph-level representation: $\phi_{\mathcal{F}}^L(G) := \mathsf{readout}\left(\{\!\{\phi_{\mathcal{F}}^{(L)}(G, v) \mid v \in V_G\}\!\}\right)$, with readout being a differentiable function. This construction defines a graph invariant. We also consider the $\mathcal{F}$-WL algorithm, as introduced by Barceló et al. (2021) as an extension of the one-dimensional Weisfeiler-Leman algorithm. The $\mathcal{F}$-WL algorithm iteratively updates vertex colors. Initially, each vertex is assigned a color:

$$\mathsf{wl}_{\mathcal{F}}^{(0)}(G, v) := (\zeta_G(v), \mathsf{Hom}(F_1, G^v), \mathsf{Hom}(F_2, G^v), \ldots).$$

At each iteration $0 \le \ell \le L$, new colors are assigned as follows:

$$\mathsf{wl}_{\mathcal{F}}^{(\ell+1)}(G, v) := (\mathsf{wl}_{\mathcal{F}}^{(\ell)}(G, v), \{\!\{\mathsf{wl}_{\mathcal{F}}^{(\ell)}(G, u) \mid u \in N_G(v)\}\!\}).$$

The final graph invariant is $\mathsf{wl}_{\mathcal{F}}^{(L)}(G) := \{\!\{\mathsf{wl}_{\mathcal{F}}^{(L)}(G, v) \mid v \in V_G\}\!\}$. This invariant can be viewed as a *color histogram* in $\mathbb{N}^c$, where $c$ is the number of distinct colors, assuming a canonical ordering on colors. When the list $\mathcal{F}$ is empty we recover the 1-WL algorithm (Weisfeiler and Leman, 1968).

---

[1] We ignore vertex features when considering homomorphisms.

**Graph encoders.** *Graph encoders* are mappings $\phi$ from the set $\mathcal{G}$ of graphs to some *embedding space* $\mathcal{Z}$, typically residing in $\mathbb{R}^k$, for $k \in \mathbb{N}$. The space $\mathcal{Z}$ is assumed to be a *metric space* for a metric $d_{\mathcal{Z}}$. Examples of graph encoders are $\mathsf{Hom}_{\mathcal{F}}$, $\mathcal{F}$-MPNNs and $\mathcal{F}$-WL, for any sequence $\mathcal{F}$ of graphs and number $L \in \mathbb{N}$ of iterations. We will develop bounds for general graph encoders.

**Wasserstein distance.** Let $\|\cdot\|$ denote the Euclidean norm in $\mathbb{R}^d$, for some $d \in \mathbb{N}$. Given two distributions $\mu$ and $\nu$ on $\mathbb{R}^d$, the *p-Wasserstein distance* between $\mu$ and $\nu$ is defined as:

$$\mathcal{W}_p(\mu, \nu) := \inf_{\pi \in \Pi(\mu, \nu)} \left( \mathbb{E}_{(x,y) \sim \pi} \|x - y\|^p \right)^{1/p},$$

where $\Pi(\mu, \nu)$ denotes the set of all couplings of $\mu$ and $\nu$, i.e., distributions $\pi$ on $\mathbb{R}^d \times \mathbb{R}^d$ with $\mu$ and $\nu$ as marginals. In what follows, we restrict our attention to the 1-Wasserstein distance.

## 4 GRAPH ENCODERS: KEY PROPERTIES

Before presenting our generalization gap bounds, we first establish crucial properties of (classes of) graph encoders that play a significant role in our analysis. In particular, we revisit the relationship between classes of graph encoders in terms of their *distinguishing power*, i.e., their ability to map distinct graphs in $\mathcal{G}$ to distinct embeddings in their embedding spaces.

**Definition 4.1.** Let $\phi : \mathcal{G} \to \mathcal{Z}_\phi$ and $\phi' : \mathcal{G} \to \mathcal{Z}_{\phi'}$ be two graph encoders. We say that $\phi$ *bounds* $\phi'$ *in distinguishing power*, denoted by $\phi \sqsubseteq \phi'$, if for any two graphs $G$ and $H$ in $\mathcal{G}$,

$$\phi'(G) \neq \phi'(H) \Rightarrow \phi(G) \neq \phi(H).$$

In other words, $\phi'$ cannot distinguish more graphs than $\phi$.

Similarly, for classes $\Phi$ and $\Phi'$ of graph encoders, we say that $\Phi$ *bounds* $\Phi'$ *in distinguishing power*, denoted by $\Phi \sqsubseteq \Phi'$, if no encoder in $\Phi'$ can distinguish more graphs than any of the encoders in $\Phi$. That is, for all $\phi' \in \Phi'$, there exists a $\phi \in \Phi$ such that $\phi \sqsubseteq \phi'$. If both $\Phi \sqsubseteq \Phi'$ and $\Phi' \sqsubseteq \Phi$ hold, then we write $\Phi \equiv \Phi'$ and say that both classes have the same distinguishing power.

From the seminal papers by Morris et al. (2019) and Xu et al. (2019), we know that $\mathsf{MPNN}(L) \equiv$ 1-$\mathsf{WL}(L)$, where the argument $L$ refers to the number of layers/iterations. Similarly, $\mathcal{F}$-$\mathsf{MPNN}(L) \equiv$ $\mathcal{F}$-$\mathsf{WL}(L)$ (Barceló et al., 2021). It is also known that $\mathsf{Hom}_{\mathcal{T}} \sqsubseteq \mathsf{MPNN}$ where $\mathcal{T}$ consists of all trees (Dell et al., 2018), and $\mathsf{Hom}_{\mathcal{T} \circ \mathcal{F}} \sqsubseteq \mathcal{F}$-$\mathsf{MPNN}(L)$ where $\mathcal{T} \circ \mathcal{F}$ consists of trees joined with copies of graphs in $\mathcal{F}$ (Barceló et al., 2021). Recent work by Neuen (2024) provides valuable insights comparing $\mathsf{Hom}_{\mathcal{F}}$ for various $\mathcal{F}$ (see also (Lanzinger and Barcelo, 2024)).

When graph encoders are comparable in terms of distinguishing power, one can recover the least expressive encoder from the most expressive one. This is formalized in the following lemma. The proofs of the results in this section can be found in Appendix C.

**Lemma 4.2.** *Let* $\phi : \mathcal{G} \to \mathcal{Z}_\phi$ *and* $\phi' : \mathcal{G} \to \mathcal{Z}_{\phi'}$ *be two graph encoders such that* $\phi \sqsubseteq \phi'$ *holds. Then there exists a function* $f : \mathcal{Z}_\phi \to \mathcal{Z}_{\phi'}$ *such that* $\phi' = f \circ \phi$.

As an illustration, consider an $L$-layer MPNN M; we know that 1-$\mathsf{WL}(L) \sqsubseteq \mathsf{M}$. It now suffices to define $f$ such that it maps a color histogram $\mathbf{h}$ to $\mathsf{M}(G)$, the embedding of $G$ by M, where $G$ is a graph satisfying $\mathsf{wl}^{(L)}(G) = \mathbf{h}$. This is well-defined due to the earlier observation that 1-$\mathsf{WL}(L) \sqsubseteq \mathsf{M}$.

For some classes of graph encoders, the function $f$ satisfies additional desirable properties, as we explain next. We say that a graph encoder $\phi : \mathcal{G} \to \mathcal{Z}_\phi$ is *B-bounded* if $d_{\mathcal{Z}_\phi}(\phi(G), \phi(H)) \leq B$ for any $G, H \in \mathcal{G}$. Furthermore, a graph encoder $\phi : \mathcal{G} \to \mathcal{Z}_\phi$ is *S-separating* if $d_{\mathcal{Z}_\phi}(\phi(G), \phi(H)) \geq S$ for any $G, H \in \mathcal{G}$ such that $\phi(G) \neq \phi(H)$. We recall that a function $f$ between metric spaces $\mathcal{Z}$ and $\mathcal{Z}'$ is *Lipschitz* with constant $\mathrm{Lip}(f)$ if for any $z_1, z_2 \in \mathcal{Z}$, $d_{\mathcal{Z}'}(f(z_1), f(z_2)) \leq \mathrm{Lip}(f) d_{\mathcal{Z}}(z_1, z_2)$. For simplicity, we set $\mathrm{Lip}(f) = \infty$ when $f$ is not Lipschitz for any finite constant.

**Proposition 4.3.** *Let* $\phi : \mathcal{G} \to \mathcal{Z}_\phi$ *be an S-separating graph encoder and* $\phi' : \mathcal{G} \to \mathcal{Z}_{\phi'}$ *be a B-bounded graph encoder such that* $\phi \sqsubseteq \phi'$. *Then* $\phi' = f \circ \phi$ *for a function* $f : \mathcal{Z}_\phi \to \mathcal{Z}_{\phi'}$ *which is Lipschitz with constant* $\mathrm{Lip}(f) = B/S$.

There are plenty of bounded graph encoders; indeed, just consider any GNN employing bounded-range activation functions such as sigmoid, tanh, truncated ReLU (Hamilton et al., 2017). Other

examples include normalized homomorphism count vectors or color histograms (Lovász and Szegedy, 2006). Similarly, any graph encoder mapping graphs into a discrete subset of $\mathbb{R}^d$ is $S$-separating. For example, any $\mathsf{Hom}_{\mathcal{F}}$ is 1-separating since whenever $\mathsf{Hom}_{\mathcal{F}}(G) \neq \mathsf{Hom}_{\mathcal{F}}(H)$, there exists an $F \in \mathcal{F}$ such that $\mathsf{Hom}(F, G) \neq \mathsf{Hom}(F, H)$. Since the latter are natural numbers, and assuming a discrete metric $d$, $d\big(\mathsf{Hom}(F, G), \mathsf{Hom}(F, H)\big) \geq 1$. A similar argument applies to graph encoders based on 1-WL or its higher-order variant $k$-WL.

Our generalization bounds use the 1-Wasserstein distance between distributions, as we will see shortly. Using Proposition 4.3, and in particular the Lipschitz property, we can relate the Wasserstein distance between the pushforward distributions of distributions of $\mathcal{G}$ on the embedding spaces of the graph encoders. Formally, let $\phi : \mathcal{G} \to \mathcal{Z}_\phi$ be a graph encoder and let $\mu$ be a distribution on $\mathcal{G}$. Then the *pushforward distribution of $\mu$ under $\phi$* is the distribution on $\mathcal{Z}_\phi$ given by

$$\phi_\sharp(\mu)(z) := \mu\big(\{G \in \mathcal{G} \mid \phi(G) = z\}\big),$$

where $z$ is an element in the embedding space $\mathcal{Z}_\phi$. We can now state the proposition.

**Proposition 4.4.** *Let $\phi : \mathcal{G} \to \mathcal{Z}_\phi$ and $\phi' : \mathcal{G} \to \mathcal{Z}_{\phi'}$ be two graph encoders such that $\phi' = f \circ \phi$. Then for any distributions $\nu$ and $\nu'$ over $\mathcal{G}$, we have that the inequality $\mathcal{W}_1\big(\phi'_\sharp(\nu), \phi'_\sharp(\nu')\big) \leq \mathrm{Lip}(f) \cdot \mathcal{W}_1\big(\phi_\sharp(\nu), \phi_\sharp(\nu')\big)$ holds.*

We remark that the inequality above becomes vacuous when $f$ is not Lipschitz and hence $\mathrm{Lip}(f) = \infty$.

**Corollary 4.5.** *Let $\phi : \mathcal{G} \to \mathcal{Z}_\phi$ be an $S$-separating graph encoder and $\phi' : \mathcal{G} \to \mathcal{Z}_{\phi'}$ a $B$-bounded graph encoder such that $\phi \sqsubseteq \phi'$ holds. Then for any distributions $\nu$ and $\nu'$ over $\mathcal{G}$, we have*

$$\mathcal{W}_1\big(\phi'_\sharp(\nu), \phi'_\sharp(\nu')\big) \leq (B/S) \cdot \mathcal{W}_1\big(\phi_\sharp(\nu), \phi_\sharp(\nu')\big).$$

Indeed, Proposition 4.3 implies $\mathrm{Lip}(f) = B/S$. Combined with Proposition 4.4, this gives $\mathcal{W}_1\big(\phi'_\sharp(\nu), \phi'_\sharp(\nu')\big) \leq \mathrm{Lip}(f) \cdot \mathcal{W}_1\big(\phi_\sharp(\nu), \phi_\sharp(\nu')\big) = (B/S) \cdot \mathcal{W}_1\big(\phi_\sharp(\nu), \phi_\sharp(\nu')\big)$.

As an example, consider a $B$-bounded graph encoder $\phi : \mathcal{G} \to \mathcal{Z}_\phi$ which is bounded in distinguishing power by the 1-separating encoder $\mathsf{Hom}_{\mathcal{F}}$, for some sequence $\mathcal{F}$ of graphs. Then for any two distributions $\nu$ and $\nu'$ on $\mathcal{G}$, we have

$$\mathcal{W}_1\big(\phi_\sharp(\nu), \phi_\sharp(\nu')\big) \leq (B/1) \cdot \mathcal{W}_1\big((\mathsf{Hom}_{\mathcal{F}})_\sharp(\nu), (\mathsf{Hom}_{\mathcal{F}})_\sharp(\nu')\big).$$

More broadly, our observations suggest that the variance of embedding distributions in $\mathcal{Z}_\phi$, produced by a complex graph encoder, can be effectively upper bounded by the variance of simpler, combinatorial graph invariants—such as homomorphism counts, Weisfeiler-Leman tests, and other structural descriptors, provided that the latter bound the former in terms of distinguishing power.

## 5    GENERALIZATION ANALYSIS

We use the setup from Chuang et al. (2021) but translated to the graph setting. More precisely, let $\mathcal{G}$ represent the input space of graphs, $\mathcal{Z}$ the embedding space in $\mathbb{R}^d$ for some $d \in \mathbb{N}$, and $\mathcal{Y} = \{1, \ldots, K\}$ the output space consisting of $K$ classes. We define a set of *graph encoders* $\Phi = \{\phi : \mathcal{G} \to \mathcal{Z}\}$ and a set of *predictors* $\Psi = \{\psi = (\psi_1, \ldots, \psi_K) : \mathcal{Z} \to \mathbb{R}^K\}$. A *score-based graph classifier* $\psi \circ \phi$ simply returns $\arg\max_{y \in \mathcal{Y}} \psi_y(\phi(G))$ on input $G$. The graph encoders in $\Phi$ are assumed to be graph invariants, such as, e.g., $\mathcal{F}$-MPNNs, $\mathcal{F}$-WL, or $\mathsf{Hom}_{\mathcal{F}}$.

We define the *margin* of a graph classifier $\psi \circ \phi$ for a graph sample $(G, y) \in \mathcal{G} \times \mathcal{Y}$ as

$$\rho_\psi(\phi(G), y) := \psi_y(\phi(G)) - \max_{y' \neq y} \psi_{y'}(\phi(G)).$$

The graph classifier $\psi \circ \phi$ misclassifies $G$ if $\rho_\psi(\phi(G), y) < 0$. Let $\mu$ be a distribution over $\mathcal{G} \times \mathcal{Y}$, and $\mathcal{S} = \{(G_i, y_i)\}_{i=1}^m$ be a set of $m$ graph samples drawn i.i.d. from $\mu$, i.e., $\mathcal{S} \sim \mu^m$. The *empirical distribution* $\mu_{\mathcal{S}}$ is defined as $\mu_{\mathcal{S}} := \frac{1}{m} \sum_{i=1}^m \delta_{(G_i, y_i)}$, where $\delta_{(G_i, y_i)}$ denotes the Dirac delta measure centered at $(G_i, y_i)$. The *expected zero-one loss* $R_\mu(\psi \circ \phi)$ and the *$\gamma$-margin empirical zero-one loss* $\hat{R}_{\gamma, \mathcal{S}}(\psi \circ \phi)$ are defined as

$$R_\mu(\psi \circ \phi) := \mathbb{E}_{(G, y) \sim \mu}\left[\mathbb{1}_{\rho_\psi(\phi(G), y) \leq 0}\right] \text{ and } \hat{R}_{\gamma, \mathcal{S}}(\psi \circ \phi) := \mathbb{E}_{(G, y) \sim \mu_{\mathcal{S}}}\left[\mathbb{1}_{\rho_\psi(\phi(G), y) \leq \gamma}\right].$$

We aim to bound the *generalisation gap* $R_\mu(\psi \circ \phi) - \hat{R}_{\gamma, \mathcal{S}}(\psi \circ \phi)$ for the graph classifier $\psi \circ \phi$.

## 5.1 GENERALIZATION BOUND

We are now ready to present the generalization bounds. Our results build on the margin bounds of Chuang et al. (2021), which are themselves based on a generalized notion of variance that involves the Wasserstein distance (Solomon et al., 2022). This notion more effectively captures the structural properties of the feature distribution. Crucially, we fully exploit the properties of graph encoders and, in particular, use Proposition 4.4 to derive an upper bound on the generalization gap of any graph encoder $\phi : \mathcal{G} \to \mathcal{Z}_\phi$ in terms of *any* graph encoder bounding $\phi$ in distinguishing power!

In order to formally state our results, some additional definitions are needed. Recall that we consider graph classifiers $(\psi_1, \ldots, \psi_K) \circ \phi$ where $\phi : \mathcal{G} \to \mathcal{Z}_\phi$ is a graph encoder and the predictor $\psi = (\psi_1, \ldots, \psi_K)$ consists of functions $\psi_i : \mathcal{Z}_\phi \to \mathbb{R}$. Recall also that the output space $\mathcal{Y} = \{1, \ldots, K\}$ and that $\mu$ is a distribution over $\mathcal{G} \times \mathcal{Y}$. We denote by $\mu_x$ the marginal distribution on $\mathcal{G}$, i.e., $\mu_x(G) := \int \mu(G, y) \mathrm{d}y$ and by $\mu_y$ the marginal distribution on $\mathcal{Y}$, i.e, $\mu_y(c) := \int \mu(x, c) \mathrm{d}x$. Then, for each $c \in \mathcal{Y}$, $\mu_c(G)$ is the conditional distribution on $\mathcal{G}$ defined by $\mu(G, c)/\mu_y(c)$.

**Theorem 5.1.** *Fix $\gamma > 0$ and a graph encoder $\phi : \mathcal{G} \to \mathcal{Z}_\phi$. Let $\lambda : \mathcal{G} \to \mathcal{Z}_\lambda$ be a graph encoder that bounds $\phi$ in distinguishing power, i.e., $\lambda \sqsubseteq \phi$. Then, for every distribution $\mu$ on $\mathcal{G} \times \mathcal{Y}$, for every predictor $\psi = (\psi_y)_{i \in \mathcal{y}}$ and every $\delta \in (0, 1)$, with probability at least $1 - \delta$ over all choices of $\mathcal{S} \sim \mu^m$, we have that the generalization gap $R_\mu(\psi \circ \phi) - \hat{R}_{\gamma, \mathcal{S}}(\psi \circ \phi)$ is upper bounded by*

$$\mathbb{E}_{c \sim \mu_y} \left[ \frac{\mathrm{Lip}\left(\rho_\psi(\cdot, c)\right) \mathrm{Lip}(f)}{\gamma} \mathbb{E}_{T, \tilde{T} \sim \mu_c^{m_c}} \left[ \mathcal{W}_1\big(\lambda_\sharp(\mu_{c,T}), \lambda_\sharp(\mu_{c,\tilde{T}})\big) \right] \right] + \sqrt{\frac{\log(1/\delta)}{2m}}, \quad (\dagger)$$

*where $\phi = f \circ \lambda$ and for each $c \in \mathcal{Y}$, $m_c$ denotes the number of pairs $(G, c)$ in $\mathcal{S}$. Also, recall that for $T \sim \mu_c^{m_c}$, $\mu_{c,T}$ is the empirical distribution $\mu_{c,T} := \sum_{G \in T} \delta_G$; similarly for $\mu_{c,\tilde{T}}$.*

The proof is a consequence of Theorem 2 in Chuang et al. (2021) and Proposition 4.4. As also observed by those authors, the expectation term over $T, \tilde{T} \sim \mu_c^{m_c}$ is intractable in general. To address this drawback, Chuang et al. (2021) show how to estimate the expectation by means of *sampling*, provided that encoders are $B$-bounded. A similar approach works in our case as well. More specifically, we show in Appendix D how an efficient, sample-based bound can be used instead of the theoretical bound presented in Theorem 5.1. Notably, this practical bound is used in our experiments.

Theorem 5.1 highlights several key factors that influence the generalization of graph classifiers: (i) the learning behavior of the predictors $\psi$, captured by $\mathrm{Lip}\left(\rho_\psi(\cdot, c)\right)$; (ii) the learning behavior of graph encoder $\phi$, relative to $\lambda$, described by $\mathrm{Lip}(f)$; and (iii) the variance of graph structures, in the Wasserstein distance $\mathcal{W}_1\big(\lambda_\sharp(\mu_{c,T}), \lambda_\sharp(\mu_{c,\tilde{T}})\big)$ for graph samples $T, \tilde{T} \sim \mu_c^{m_c}$.

## 5.2 CONCENTRATION AND SEPARATION

In terms of concentration, since $\mathbb{E}_{T, \tilde{T} \sim \mu_c^{m_c}}[\mathcal{W}_1(\lambda_\sharp(\mu_{c,T}), \lambda_\sharp(\mu_{c,\tilde{T}}))] \leq O(m^{-1/d})$ (Chuang et al., 2021), a large sample size $m$ and a small dimension $d$ of the embedding space $\mathcal{Z}_\lambda$ lead to a smaller generalization bound. For instance, when $\mu$ is concentrated on graphs in $\mathcal{G}$ with low *color complexity* (Morris et al., 2023)—i.e., the 1-WL test requires only a small number of colors for the graph's vertices—combinatorial graph encoders like $\mathrm{Hom}_\mathcal{F}$ and $\mathcal{F}\text{-WL}(L)$ can operate in low-dimensional spaces. This observation is consistent with earlier findings (Kiefer and McKay, 2020; Garg et al., 2020; Liao et al., 2021; Ju et al., 2023; Cong et al., 2021; Esser et al., 2021; Morris et al., 2023) about the effect of graph size, degree, and maximum degree on generalization performance.

Of particular interest is the case when the bounding graph classifier $\lambda$ is assumed to have a large margin. A larger margin is generally associated with better generalization (Elsayed et al., 2018; Chuang et al., 2021). If we assume the margin $\gamma$ is satisfied for $\psi \circ \lambda$, for all graph samples, and for each $c \in \mathcal{Y}$, the predictor $\psi_c \in \psi$ is Lipschitz, then (see Lemma 10 in Chuang et al. (2021)) we have

$$\gamma \leq \big( \min_{\substack{c,c' \in \mathcal{Y} \\ c \neq c'}} \mathcal{W}_1(\lambda_\sharp(\mu_c), \lambda_\sharp(\mu_{c'})) \big) \big( \min_{c \in \mathcal{Y}} \mathrm{Lip}(\psi_c) \big).$$

By replacing $1/\gamma$ in Equation ($\dagger$) by this bound, we obtain Proposition 5.2, see Appendix D for details. We hereby revealing a trade-off between concentration and separation.

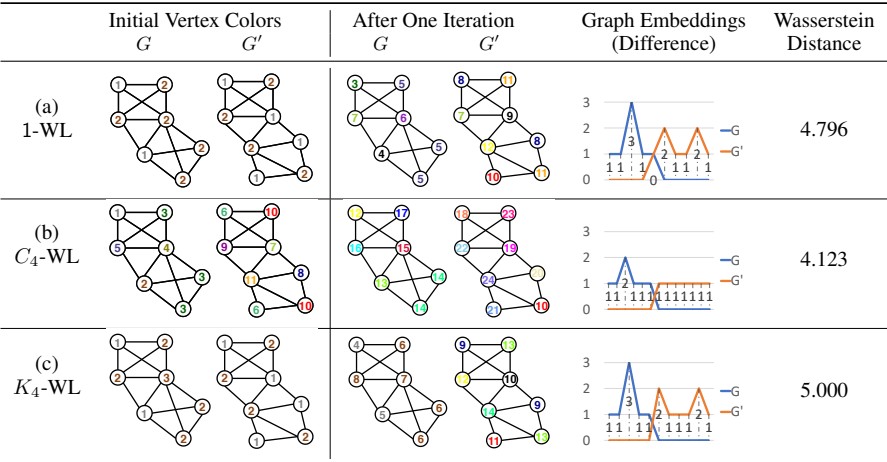

Table 1: Two graphs $G$ and $G'$ with the initial vertex colors (including input vertex features and homomorphism counts) and the vertex colors after one iteration, and the dimension-wise difference $|\lambda_\sharp(G) - \lambda_\sharp(G')|$ and Wasserstein distance $\mathcal{W}_1\big(\lambda_\sharp(G), \lambda_\sharp(G')\big)$ between their graph embeddings for three models: (a) 1-WL, (b) $C_4$-WL, and (c) $K_4$-WL.

**Proposition 5.2.** *Under the same assumptions as in Theorem 5.1, but with the additional requirement that the predictors $\psi_c$ in $\psi$ are Lipschitz, and that the bounding graph classifier $\lambda$ has a large margin, i.e., $\rho_\psi(\lambda(G), y) \geq \gamma$ for all $(G, y) \sim \mu$, then for any $\delta \in (0, 1)$, with probability at least $1 - \delta$ over all choices $\mathcal{S} \sim \mu^m$, we have that the generalization bound given in Theorem 5.1 is lower bound by*

$$\frac{\mathsf{Lip}(f) \cdot \mathbb{E}_{c \sim \mu_y}\left[\mathsf{Lip}(\rho_\psi(\cdot, c))\mathbb{E}_{T, \tilde{T} \sim \mu_c^{m_c}}\left[\mathcal{W}_1\left(\lambda_\sharp(\mu_{c,T}), \lambda_\sharp(\mu_{c,\tilde{T}})\right)\right]\right]}{\left(\min_{c \in \mathcal{Y}} \mathsf{Lip}(\psi_c)\right)\left(\min_{c,c' \in \mathcal{Y}, c \neq c'} \mathcal{W}_1\left(\lambda_\sharp(\mu_c), \lambda_\sharp(\mu_{c'})\right)\right)} + \sqrt{\frac{\log(1/\delta)}{2m}}.$$

The above proposition highlights that, to achieve a low generalization bound, it is crucial to ensure good concentration between embeddings of the same class, i.e., $\mathcal{W}_1(\lambda_\sharp(\mu_{c,T}), \lambda_\sharp(\mu_{c,\tilde{T}}))$, while maintaining a large separation between embeddings of different classes, i.e., $\mathcal{W}_1(\lambda_\sharp(\mu_c), \lambda_\sharp(\mu_{c'}))$, where $c \neq c'$. This can be achieved when $\lambda$ learn embeddings in the "right" directions, where embeddings of different classes are "more separated" than those of the same class, or when the distribution $\mu$ is concentrated on graphs for which this separation happens for $\lambda$.

**Remarks.** In our bounds, we identify three Lipschitz constants: $\mathsf{Lip}(\rho_\psi(\cdot, c))$, $\mathsf{Lip}(\psi_c)$, and $\mathsf{Lip}(f)$. First, note that $\rho_\psi(\cdot, c)$ depends on $\psi_c$, and therefore it is Lipschitz in its first argument if $\psi_c$ is Lipschitz. For simplicity, we assume that the predictor $\psi = (\psi_c)_{c \in \mathcal{Y}}$ is a softmax function with Lipschitz constant 1. For general $\psi_c$, $\mathsf{Lip}(\rho_\psi(\cdot, c))$ can be approximated empirically using the Jacobian, as suggested by (Chuang et al., 2021).

Furthermore, Corollary 4.5 states that the connecting function $f$ between the graph encoders $\lambda \sqsubseteq \phi$ is Lipschitz with constant $B/S$, provided that $\phi$ is $B$-bounded and $\lambda$ is separating. Therefore, when $\phi$ is $B$-bounded, $\mathsf{Lip}(f)$ decreases as $S$ increases. We also note that $S$ can increase with added expressivity in $\lambda$, which enhances its separation ability. In practice, both $B$ and $S$ can be computed empirically. We discuss the effect of added expressivity of $\lambda$ in more detail in the next section.

## 6 CASE STUDIES

In this section, we present case studies to illustrate how our generalization bound captures complex scenarios in the generalization of graph encoders, influenced by their model expressivity and driven by two key factors: intra-class concentration and inter-class separation. Recall, as discussed in Theorem 5.1 and Proposition 5.2: (i) *intra-class concentration*, which quantifies the variance of graph structures within a class, measured by the Wasserstein distance $\mathcal{W}_1(\lambda_\sharp(\mu_{c,T}), \lambda_\sharp(\mu_{c,\tilde{T}}))$ for graph

samples $T, \tilde{T} \sim \mu_c^{m_c}$, and (ii) *inter-class separation*, which measures the distinction between classes, represented by the Wasserstein distance $\max_{c,c' \in \mathcal{Y}, c \neq c'} \mathcal{W}_1(\lambda_\sharp(\mu_c), \lambda_\sharp(\mu_{c'}))$.

For simplicity, we consider the following graphs $\{G, G', H, H'\}$ from the PROTEINS dataset (Morris et al., 2020a), uniformly selected by the distribution $\mu$,

$$ G = \quad\quad\quad G' = \quad\quad\quad H = \quad\quad\quad H' = $$

Here, $G$ and $G'$ belong to the class $c$, while $H$ and $H'$ belong to the class $c'$, where $c \neq c'$. Assuming the margin condition is satisfied for all classes, including $c$ and $c'$, and that embeddings of graphs within each class cluster around the embeddings of $G$, $G'$, and $H$, $H'$, we estimate $\mathcal{W}_1(\lambda_\sharp(\mu_{c,T}), \lambda_\sharp(\mu_{c,\tilde{T}}))$ using $\mathcal{W}_1(\lambda_\sharp(G), \lambda_\sharp(G'))$. Since there are only two classes in this dataset, we estimate $\max_{c,c' \in \mathcal{Y}, c \neq c'} \mathcal{W}_1(\lambda_\sharp(\mu_c), \lambda_\sharp(\mu_{c'}))$ by $\mathcal{W}_1\big(\lambda_\sharp(\{G, G'\}), \lambda_\sharp(\{H, H'\})\big)$.

For the bounding graph encoders $\lambda$, we use 1-WL as the base model. Then, following the approach by Barceló et al. (2021) we consider two simple rooted graphs and and append their homomorphism counts $\mathsf{Hom}(, \cdot)$ and $\mathsf{Hom}(, \cdot)$ to the vertex feature of each rooted pair $(\cdot, v)$ in the graphs $G$, $G'$, $H$ and $H'$, respectively. This leads to slight increase in model expressivity, compared with 1-WL, allowing us to analyze how these differences impact key factors in generalization. We refer to these two graph encoders as $C_4$-WL, where homomorphism counts of are added, and $K_4$-WL, where homomorphism counts of are added, respectively. Table 1 presents graphs $G$ and $G'$ with their initial vertex colors and the updated colors after one iteration, where the Wasserstein distance $\mathcal{W}_1(\lambda_\sharp(G), \lambda_\sharp(G'))$ estimates the intra-class concentration of class $c$. As model expressivity increases from 1-WL to $C_4$-WL and $K_4$-WL, two distinct scenarios for intra-class concentration arise:

(1) *More expressivity leads to better generalization*: Compared to the graph embeddings from 1-WL, incorporating $C_4$ improves intra-class concentration. The homomorphism counts of $C_4$ reduce the variance between graph embeddings as shown in Table 1, resulting in a distance of 4.123, smaller than 4.796 for 1-WL.

(2) *More expressivity leads to worse generalization*: When $K_4$ is used, the graph embeddings of $G$ and $G'$ yield a distance of 5.000, which is larger than its 1-WL counterpart 4.796. Compared to $C_4$-WL, each dimension of the $K_4$-WL embeddings has the same or larger magnitude, reflecting higher variance in the graph embeddings.

When measuring inter-class separation using $\mathcal{W}_1(\lambda_\sharp(\mu_c), \lambda_\sharp(\mu_{c'}))$, the models 1-WL, $C_4$-WL, and $K_4$-WL achieve distances of 4.582, 4.511, and 4.840, respectively. These results suggest a narrowing in the gaps of these models, compared to intra-class concentration alone. The trends in inter-class separation may change depending on the graph structure. For instance, if graphs of class $c'$ cluster around the embedding of $H'$, i.e., estimating $\mathcal{W}_1(\lambda_\sharp(\mu_c), \lambda_\sharp(\mu_{c'}))$ with $\mu_c = \{G, G'\}$ and $\mu_{c'} = \{H'\}$, the reverse trend may occur, with 1-WL achieving a distance of 4.796 and $K_4$-WL achieving 4.583. This highlights the importance of inter-class separation in balancing a model's generalization performance alongside intra-class concentration.

## 7 EXPERIMENTS

**Tasks and Datasets**    We conduct graph classification experiments on six widely used benchmark datasets: ENZYMES, PROTEINS, and MUTAG from the TU dataset collection (Morris et al., 2020a), as well as SIDER and BACE from the molecular dataset collection (Wu et al., 2017). For SIDER, which comprises 27 classification tasks, we focus specifically on the 21st task. Each dataset is randomly divided into training and test sets following a 90%/10% split.

**Setup and Configuration**    Each classification task is trained for 400 epochs, with five independent runs to report the mean and standard deviation of the results. Consistent with the setup in Tang and Liu (2023); Morris et al. (2023); Cong et al. (2021), we eliminate the use of regularization techniques such as dropout and weight decay. A batch size of 128 is utilized, with a learning rate set to $10^{-3}$, and the hidden layer dimension fixed at 64. The margin loss function is employed with a margin parameter $\gamma = 1$. To compute the generalization gap, we utilize the sample-based variant of the bound as

Table 2: Left: Graph classification gaps and bounds with different numbers of MPNN layers. Right: Correlation matrices of empirical gaps and bounds.

| # Layers | | ENZYMES | PROTEINS | Dataset MUTAG | SIDER | BACE |
|---|---|---|---|---|---|---|
| 1 | Loss gap | $0.248_{\pm0.040}$ | $0.029_{\pm0.015}$ | $-0.070_{\pm0.017}$ | $0.037_{\pm0.003}$ | $0.018_{\pm0.017}$ |
| | Our Bound | $7.926_{\pm1.279}$ | $2.193_{\pm0.702}$ | $1.216_{\pm0.169}$ | $0.511_{\pm0.286}$ | $1.479_{\pm0.301}$ |
| | VC dimension | 586 | 929 | 51 | 960 | 621 |
| | VC bound | $1.302_{\pm0.000}$ | $1.292_{\pm0.001}$ | $1.100_{\pm0.004}$ | $1.302_{\pm0.000}$ | $1.301_{\pm0.000}$ |
| | PAC bound | 3.48 | 5.04 | 3.06 | 52.39 | 21.525 |
| 1 | Loss gap | $0.242_{\pm0.026}$ | $0.032_{\pm0.010}$ | $-0.074_{\pm0.007}$ | $0.038_{\pm0.003}$ | $0.037_{\pm0.019}$ |
| | Our bound | $7.425_{\pm0.982}$ | $1.404_{\pm0.144}$ | $1.247_{\pm0.155}$ | $0.620_{\pm0.463}$ | $1.729_{\pm0.251}$ |
| | VC dimension | 595 | 996 | 121 | 1300 | 1060 |
| | VC bound | $1.302_{\pm0.000}$ | $1.292_{\pm0.000}$ | $1.281_{\pm0.003}$ | $1.302_{\pm0.000}$ | $1.302_{\pm0.000}$ |
| | PAC bound | 12.75 | 31.94 | 8.17 | $132.79_{\pm8.12}$ | 51.573 |
| 2 | Loss gap | $0.237_{\pm0.035}$ | $0.025_{\pm0.009}$ | $-0.058_{\pm0.012}$ | $0.038_{\pm0.002}$ | $0.032_{\pm0.011}$ |
| | Our Bound | $6.513_{\pm0.951}$ | $1.421_{\pm0.220}$ | $1.649_{\pm0.158}$ | $0.409_{\pm0.253}$ | $1.789_{\pm0.226}$ |
| | VC dimension | 595 | 996 | 135 | 1309 | 1089 |
| | VC bound | $1.302_{\pm0.000}$ | $1.293_{\pm0.000}$ | $1.286_{\pm0.002}$ | $1.302_{\pm0.000}$ | $1.302_{\pm0.000}$ |
| | PAC bound | 56.98 | 276.78 | $21.96_{\pm0.00}$ | 341.04 | 124.605 |
| 4 | Loss gap | $0.235_{\pm0.038}$ | $0.027_{\pm0.005}$ | $-0.073_{\pm0.009}$ | $0.036_{\pm0.001}$ | $0.022_{\pm0.030}$ |
| | Our Bound | $6.825_{\pm0.796}$ | $1.434_{\pm0.297}$ | $1.535_{\pm0.115}$ | $0.298_{\pm0.080}$ | $1.686_{\pm0.377}$ |
| | VC dimension | 595 | 996 | 139 | 1309 | 1093 |
| | VC bound | $1.302_{\pm0.000}$ | $1.292_{\pm0.001}$ | $1.291_{\pm0.002}$ | $1.302_{\pm0.000}$ | $1.302_{\pm0.000}$ |
| | PAC bound | 308.43 | 2331.63 | 57.69 | 845.62 | 310.732 |
| 5 | Loss gap | $0.256_{\pm0.037}$ | $0.020_{\pm0.007}$ | $-0.071_{\pm0.021}$ | $0.035_{\pm0.001}$ | $0.020_{\pm0.020}$ |
| | Our Bound | $6.384_{\pm0.813}$ | $1.308_{\pm0.165}$ | $1.773_{\pm0.194}$ | $0.369_{\pm0.172}$ | $1.662_{\pm0.120}$ |
| | VC dimension | 595 | 996 | 139 | 1309 | 1093 |
| | VC bound | $1.302_{\pm0.000}$ | $1.292_{\pm0.001}$ | $1.292_{\pm0.002}$ | $1.302_{\pm0.000}$ | $1.302_{\pm0.000}$ |
| | PAC bound | 1615.10 | 17992.81 | 155.74 | 2179.21 | 744.08 |
| 6 | Loss gap | $0.264_{\pm0.025}$ | $0.030_{\pm0.008}$ | $-0.078_{\pm0.019}$ | $0.034_{\pm0.002}$ | $0.022_{\pm0.016}$ |
| | Our Bound | $6.151_{\pm0.798}$ | $1.340_{\pm0.316}$ | $1.627_{\pm0.038}$ | $0.353_{\pm0.156}$ | $1.785_{\pm0.237}$ |
| | VC dimension | 595 | 996 | 139 | 1309 | 1093 |
| | VC bound | $1.302_{\pm0.000}$ | $1.292_{\pm0.001}$ | $1.291_{\pm0.002}$ | $1.302_{\pm0.000}$ | $1.302_{\pm5.870}$ |
| | PAC bound | 8931.00 | 135762.52 | 410.31 | 5254.88 | 1860.94 |

outlined in Theorem 5.1, as given in Theorem D.2 of the appendix. For the graph encoder $\phi$, we adopt both MPNNs and $\mathcal{F}$-MPNNs, with expressivity constraints defined by 1-WL and $\mathcal{F}$-WL, respectively, as described in Section 4. The predictor $\psi(\cdot)$ is modeled using the softmax function, which has a Lipschitz constant of 1 (Gao and Pavel, 2017), ensuring that $\mathsf{Lip}(\rho_\psi(\cdot, c))$ is also 1. We estimate $\mathsf{Lip}(f)$ as: $\mathsf{Lip}(f) = \max_{G, H \in \mathcal{G}_{\text{train}}} \left( \frac{d_{\mathcal{Z}_\phi}(\phi(G), \phi(H))}{d_{\mathcal{Z}_\lambda}(\lambda(G), \lambda(H))} \right)$, where $G$ and $H$ are sampled from the training set $\mathcal{G}_{\text{train}}$. For all experiments, we set the confidence level $\delta$ to 0.1, yielding bounds with high probability.

## 7.1 RESULTS AND DISCUSSION

**How well can the proposed bound predict the generalization ability of MPNNs?** We compare our bound with empirical generalization gaps, measured by loss, while varying the number of layers (see Table 2). For comparison, we include the VC bound from Morris et al. (2023), based on unique color histograms (VC dimension) from 1-WL, and the PAC-Bayesian bound from Ju et al. (2023). Changes in loss gaps are shown in Figure 2 of Appendix F. Our bound strongly correlates with empirical gaps across datasets and depths, effectively predicting generalization errors and reflecting performance changes as model depth increases. In contrast, the VC bound stabilizes after three layers and remains nearly constant, failing to capture the observed variations. Furthermore, our bound surpasses the PAC-Bayesian bound in both tightness and correlation to empirical gaps, notably on deeper MPNNs, since the PAC-Bayesian bound grows exponentially with the number of layers. Similarly, our bound is less vacuous compared to other bounds, such as those proposed by Garg et al. (2020); Liao et al. (2021), which tend to be on the order of $10^4$.

We further evaluate $\mathcal{F}$-MPNNs across different homomorphism patterns, as shown in Figure 1. Here, $P_n$, $K_n$, and $C_n$ denote $n$-path, $n$-clique, and $n$-cycle graphs, respectively, and "no pattern" refers to the MPNN without a specific pattern. Overall, the generalization bound aligns with the empirical gap across patterns, with some exceptions in ENZYMES. In ENZYMES, cycle patterns yield a larger gap than cliques or paths, whereas in PROTEINS, paths and cliques increase the gap and cycles reduce it. These variations are largely captured by the corresponding bounds.

**Why does more expressive power sometimes lead to better generalization?** In Figure 1, we observe two contrasting cases where increased expressivity worsens generalization (ENZYMES) and

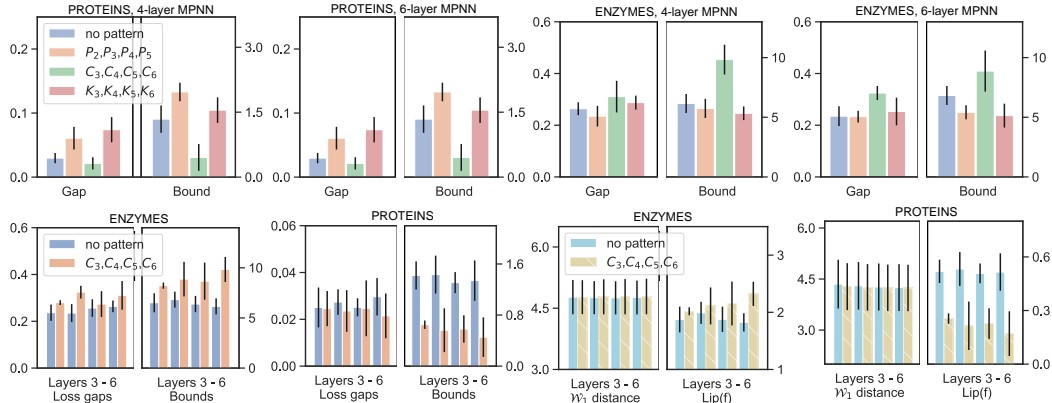

Figure 1: Top: Loss gaps and bounds of different patterns. Bottom: Loss gaps, bounds, Wasserstein distance and $\text{Lip}(f)$ of different layers. The data sets used are PROTEINS and ENZYMES.

improves it (PROTEINS). To explore this further, we plot the changes of two major factors from the proposed bound in Theorem D.2: the 1-Wasserstein distance and $\text{Lip}(f)$, both shown in Figure 1. The 1-Wasserstein distance ($\mathcal{W}_1$) is computed as: $\frac{1}{n}\sum_{j=1}^{n}\mathcal{W}_1\big(\lambda_\sharp(\mu_{c,T^j}),\lambda_\sharp(\mu_{c,\tilde{T}^j})\big)$ averaged over all graph classes. We plot these factors over four layers for both MPNN and $\{C_3, C_4, C_5, C_6\}$-MPNN. We observe that the inclusion of homomorphism counts worsens generalization in ENZYMES but improves it in PROTEINS. This can be attributed to the joint influence of the Wasserstein distance and $\text{Lip}(f)$. In ENZYMES, both the 1-Wasserstein distance and $\text{Lip}(f)$ increase slightly when homomorphism counts are added. While this additional expressivity leads to better separation between graphs, in ENZYMES, this increased separation hinders the ability to achieve good concentration within each graph class, ultimately worsening generalization. In contrast, for PROTEINS, although the inclusion of homomorphism counts leads to greater graph separation, it also slightly reduces the 1-Wasserstein distance within each class, allowing for better concentration. This improved separation significantly reduces $\text{Lip}(f)$, resulting in enhanced generalization.

**Can generalization be improved by controlling the Lipschitz constants?** Last but not least, since $\text{Lip}(f)$ plays a crucial role in the proposed bound, we aim to investigate whether controlling $\text{Lip}(f)$ can serve as an effective strategy to enhance generalization. A straightforward approach to control $\text{Lip}(f)$ is through normalization techniques. As demonstrated earlier, normalization effectively bounds the diameter of $\phi_\sharp(\mu)$, which, in turn, constrains the encoder's boundedness and subsequently $\text{Lip}(f)$. To test this, we apply $l_1$-normalisation in the last layer of the MPNN. See Table 3 for results. It is evident that normalization reduces the generalization gap across all datasets. This improvement is also reflected in the computed bounds. Interestingly, the least improvement is observed in the SIDER dataset, where $\text{Lip}(f)$ is already relatively small, and the embeddings are well-concentrated even before normalization. This suggests that the impact of normalization is more pronounced when $\text{Lip}(f)$ is large or when the embeddings are not already well-concentrated.

## 8    CONCLUSION AND LIMITATIONS

In this work, we examine the generalization of GNNs from a margin-based perspective, based on the work by Chuang et al. (2021). The bounds use 1-variance and optimal transport to analyze graph embeddings. We establish a relationship between generalization and the expressive capacity of GNNs, deriving a generalization bound that demonstrates how well-clustered embeddings and separable classes lead to improved generalization. Through case studies on a real-world dataset, we empirically validate these theoretical findings. We also apply empirical sample-based bounds to graph classification tasks, confirming that our theoretical results align with empirical evidence. Our work enables analyzing the generalization of graph encoders through their bounded expressive power.

Nonetheless, our work has some limitations. While we validate the framework on real-world datasets, further large-scale studies across a wider range of datasets and applications are needed to fully establish the proposed approach's general applicability.

ACKNOWLEDGEMENTS

This research was supported partially by the Australian Government through the Australian Research Council's Discovery Projects funding scheme (project DP210102273) and by the National Research Foundation of Korea (NRF) grant funded by the Korea government(MSIT)(RS-2024-00337955 and RS-202300217286).

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

## A    ADDITIONAL RELATED WORK

We provide additional references related to the expressiveness of Graph Neural Networks (GNNs). The connection with the Weisfeiler-Leman (1-WL) test has led to the development of high-order GNNs that surpass 1-WL and are bounded by the $k$-dimensional Weisfeiler-Leman test ($k$-WL) (Morris et al., 2019; Maron et al., 2019; Morris et al., 2020b; Geerts and Reutter, 2022). The method by Morris et al. (2019; 2020b) is strictly weaker than $k$-WL, whereas the method by Maron et al. (2019) can match the expressiveness of $k$-WL. However, these higher-order GNNs incur significant computational costs, rendering them impractical for large-scale datasets.

Incorporating substructure counts has been shown to be an effective strategy for enhancing GNN expressivity beyond 1-WL (Bouritsas et al., 2023; Barceló et al., 2021). Bouritsas et al. (2023) integrate isomorphism counts of small subgraph patterns into the node and edge features of graphs, while Barceló et al. (2021) employ a similar approach using homomorphism counts. Building on this concept, Thiede et al. (2021) implemented convolutions on automorphism groups of subgraph patterns. Rather than directly using subgraph counts, Wijesinghe and Wang (2022); Wang et al. (2023) propose integrating local structural information into neighbor aggregation. This approach suggests that the expressivity of the model increases with the subgraph pattern size and aggregation radius.

Taking a different approach, Nguyen and Maehara (2020) explore the use of graph homomorphism counts directly in convolutions without message passing, demonstrating their universality in approximating invariant functions. Welke et al. (2023) propose combining homomorphism counts with GNN outputs in the final layer to improve expressivity. Additionally, Bevilacqua et al. (2022) represent graphs as collections of subgraphs derived from a predetermined policy. Zhao et al. (2022) and Zhang and Li (2021) extend this idea by representing graphs with a set of induced subgraphs. These methods are closely related to graph kernel techniques that utilize subgraph patterns (Shervashidze et al., 2011; Horváth et al., 2004; Costa and Grave, 2010).

Since the WL-based GNN expressivity hierarchy is inherently coarse and qualitative, Zhang et al. (2024) propose a homomorphism-based expressivity framework, which enables direct comparisons of expressivity between common GNN models. As 1-WL and $k$-WL have equivalent translations in homomorphism embeddings (Dell et al., 2018), both MPNNs and higher-order GNNs can be expressed using homomorphism representations within this framework. Given that homomorphism embeddings are theoretically isomorphism-complete, this framework offers not only a unified but also a complete description of GNN expressivity.

# B    ADDITIONAL DETAILS OF SECTION 3

We provide some examples illustrating the key concepts introduced in Section 3.

**Example B.1** (Homomorphism counts and graph invariants). Consider the following three graphs $F_1$, $F_2$ and $G$:

$$F_1 = \quad\quad F_2 = \quad\quad G =$$

Suppose that we want to extract graph features from $G$ based on the graph patterns $F_1$ and $F_2$. One way of doing so is by means of counting how many homomorphisms from the patterns to $G$ exist. We recall that a homomorphism is just an edge-preserving mapping between the vertex sets. For example, here, one can verify that the number of homomorphisms from $F_1$ to $G$ is 120, i.e., $\mathsf{Hom}(F_1, G) = 120$, and the number of homomorphisms from $F_2$ to $G$ is 8, i.e., $\mathsf{Hom}(F_2, G) = 8$. These homomorphism counts can be used as features to enrich the data representation of $G$. The process maps $G$ to a set of numerical features derived from the counts. Importantly, this mapping is a graph invariant, meaning that if $G$ is replaced by an isomorphic graph (one structurally identical to $G$), the homomorphism counts remain the same. Using homomorphism counts as features allows us to capture structural information, making them valuable for tasks like classification or regression in graph-based machine learning.

**Example B.2** (Rooted homomorphism counts and vertex invariants). Now consider the following three rooted graphs $F_1^r, F_2^r, G^v$:

$$F_1^r = \quad\quad F_2^r = \quad\quad G^v =$$

The roots in the graph allow to connect homomorphism counts locally around each vertex. Indeed, for rooted graphs, the homomorphisms also have to preserve the roots. In this example one can verify that the number of homomorphisms from $F_1^r$ to $G^v$ is 78, i.e., $\mathsf{Hom}(F_1^r, G^v) = 78$, and the number of homomorphisms from $F_2^r$ to $G^v$ is 4, i.e., $\mathsf{Hom}(F_2^r, G^v) = 4$. We can enrich the local graph structure around $v$ in this way. The mapping that associates with graphs and vertices such rooted homomorphism counts is an example of an vertex invariant.

**Example B.3** (Wasserstein distance). Finally we illustrate the notion of Wasserstein distance. Consider the four graphs presented in Section 6:

$$G = \quad\quad G' = \quad\quad H = \quad\quad H' =$$

The vector representations of the four graphs after one iteration of 1-WL are

$$\lambda(G) = (1, 0, 3, 1, 0, 0, 0, 0, 0, 1, 0, 0, 0, 1),$$
$$\lambda(G') = (0, 1, 0, 0, 1, 1, 0, 0, 2, 1, 0, 2, 0, 0),$$
$$\lambda(H) = (1, 0, 3, 1, 0, 0, 0, 0, 0, 1, 0, 0, 0, 1),$$
$$\lambda(H') = (0, 1, 0, 0, 1, 1, 0, 0, 2, 1, 0, 2, 0, 0).$$

These lead to

$$||\lambda(G) - \lambda(H)|| = 5.0990,$$
$$||\lambda(G') - \lambda(H')|| = 4.7958,$$
$$||\lambda(G) - \lambda(H')|| = 5.2915,$$
$$||\lambda(G') - \lambda(H)|| = 3.8730.$$

Let $\mu = \{G, G'\}$ and $\nu = \{H, H'\}$. Then $\Pi(\mu, \nu) = \{\{(G, H), (G', H')\}, \{(G, H'), (G', H)\}\}$. Thus, we have

$$\mathbb{E}_{(x,y)\sim\{(G,H),(G',H')\}}||x - y|| = \frac{1}{2}\left(||\lambda(G) - \lambda(H)|| + ||\lambda(G') - \lambda(H')||\right)$$
$$= 4.9474;$$

$$\mathbb{E}_{(x,y)\sim\{(G,H'),(G',H)\}}||x - y|| = \frac{1}{2}\left(||\lambda(G) - \lambda(H')|| + ||\lambda(G) - \lambda(H')||\right)$$
$$= 4.5823.$$

Since $4.5823 < 4.9474$, we obtain

$$
\begin{aligned}
\mathcal{W}(\mu, \nu) &= \mathbb{E}_{(x,y) \sim \{(G,H'),(G',H)\}} \|x - y\| \\
&= \frac{1}{2} \big( \|\lambda(G) - \lambda(H')\| + \|\lambda(G) - \lambda(H')\| \big) \\
&= 4.5823.
\end{aligned}
$$

## C  PROOFS OF SECTION 4

**Lemma 4.2.** *Let $\phi : \mathcal{G} \to \mathcal{Z}_\phi$ and $\phi' : \mathcal{G} \to \mathcal{Z}_{\phi'}$ be two graph encoders such that $\phi \sqsubseteq \phi'$ holds. Then there exists a function $f : \mathcal{Z}_\phi \to \mathcal{Z}_{\phi'}$ such that $\phi' = f \circ \phi$.*

*Proof.* We define the function $f : \mathcal{Z}_\phi \to \mathcal{Z}_{\phi'}$, as follows. Let $z \in \mathcal{Z}_\phi$ and $G \in \mathcal{G}$ such that $\phi(G) = z$. Then, define $f(z) := \phi'(G) \in \mathcal{Z}_{\phi'}$. Observe first that $f$ is well-defined. Indeed, if we take another $G' \in \mathcal{G}$ such that $\phi(G') = z$, then $\phi(G) = \phi(G')$ and hence also $\phi'(G) = \phi'(G') = f(z)$ since $\phi \sqsubseteq \phi'$ by assumption. Clearly, $f \circ \phi = \phi'$, by definition $\qquad\square$

**Proposition 4.3.** *Let $\phi : \mathcal{G} \to \mathcal{Z}_\phi$ be an S-separating graph encoder and $\phi' : \mathcal{G} \to \mathcal{Z}_{\phi'}$ be a B-bounded graph encoder such that $\phi \sqsubseteq \phi'$. Then $\phi' = f \circ \phi$ for a function $f : \mathcal{Z}_\phi \to \mathcal{Z}_{\phi'}$ which is Lipschitz with constant $\mathrm{Lip}(f) = B/S$.*

*Proof.* We need to show that for any $z, z' \in \mathcal{Z}_\phi$, $d_{\mathcal{Z}_{\phi'}}(f(z), f(z')) \leq (B/S) \cdot d_{\mathcal{Z}_\phi}(z, z')$ holds. Clearly, if $z = z'$ then also $f(z) = f(z')$ and hence $d_{\mathcal{Z}_{\phi'}}(f(z), f(z')) = 0$, for which the desired inequality trivially holds. For $z \neq z'$ and using that $z = \phi(G)$ and $z' = \phi(H)$ for some graphs $G$ and $H$ in $\mathcal{G}$, we know that $S \leq d_{\mathcal{Z}_\phi}(z, z')$ and hence $1 \leq (1/S) \cdot d_{\mathcal{Z}_\phi}(z, z')$. It now suffices to observe that $d_{\mathcal{Z}_{\phi'}}(f(z), f(z')) = d_{\mathcal{Z}_{\phi'}}(f(\phi(G)), f(\phi(H))) = d_{\mathcal{Z}_{\phi'}}(\phi'(G), \phi'(H)) \leq B$, from which $d_{\mathcal{Z}_{\phi'}}(f(z), f(z')) \leq (B/S) \cdot d_{\mathcal{Z}_\phi}(z, z')$ follows. $\qquad\square$

**Proposition 4.4.** *Let $\phi : \mathcal{G} \to \mathcal{Z}_\phi$ and $\phi' : \mathcal{G} \to \mathcal{Z}_{\phi'}$ be two graph encoders such that $\phi' = f \circ \phi$. Then for any distributions $\nu$ and $\nu'$ over $\mathcal{G}$, we have that the inequality $\mathcal{W}_1\big(\phi'_\sharp(\nu), \phi'_\sharp(\nu')\big) \leq \mathrm{Lip}(f) \cdot \mathcal{W}_1\big(\phi_\sharp(\nu), \phi_\sharp(\nu')\big)$ holds.*

*Proof.* We first show that $f \circ \phi = \phi'$ implies the $f_\sharp\big(\phi_\sharp(\mu)\big) = \phi'_\sharp(\mu)$ of the corresponding pushforward distribution of any distribution $\mu$ om $\mathcal{G}$. Indeed, this simply follows from the definitions. One the one hand, for $I \subseteq \mathcal{Z}_{\phi'}$

$$
\phi'_\sharp(\mu)(I) := \mu\big(\{G \in \mathcal{G} \mid \phi'(G) \in I\}\big).
$$

On the other hand,

$$
\begin{aligned}
f_\sharp\big(\phi_\sharp(\mu)\big)(I) &= \phi_\sharp(\mu)\big(\{z \in \mathcal{Z}_\phi \mid f(z) \in I\}\big) \\
&= \mu\big(G \in \mathcal{G} \mid f(\phi(G)) \in I\big).
\end{aligned}
$$

The equality then follows from $f \circ \phi = \phi'$. We assume that $f$ is Lipschitz-continuous with $\mathrm{Lip}(f) < \infty$ (otherwise the inequality is satisfied by default and there is nothing to prove). We show that

$$
\mathcal{W}_1\big(\phi'_\sharp(\mu), \phi'_\sharp(\nu)\big) \leq \mathrm{Lip}(f) \cdot \mathcal{W}_1\big(\phi_\sharp(\mu), \phi_\sharp(\nu)\big).
$$

Let $L_1(\mathcal{Z}_\phi)$ be the set of 1-Lipschitz functions on $\mathcal{Z}_\phi$. We use the Kantorovich-Rubinstein dual form of $\mathcal{W}_1$, as follows:

$$
\begin{aligned}
\mathcal{W}_1\big(\phi_\sharp(\mu), \phi_\sharp(\nu)\big) &= \sup_{g \in L_1(\mathcal{Z}_\phi)} \mathbb{E}_{z \sim \phi_\sharp(\mu)}[g(z)] - \mathbb{E}_{z \sim \phi_\sharp(\nu)}[g(z)] \\
&= \sup_{g \in L_1(\mathcal{Z}_\phi)} \int_{\mathcal{Z}_\phi} g(z) \, \mathrm{d}(\phi_\sharp(\mu) - \phi_\sharp(\nu))(z).
\end{aligned}
$$

Note that if $g \in L_1(\mathcal{Z})$ then $\frac{1}{\mathrm{Lip}(f)} f \circ g \in L_1(\mathcal{Z})$ as well. Then, using our earlier observation about pushforward distributions,

$$
\mathcal{W}_1\big(\phi'_\sharp(\mu), \phi'_\sharp(\nu)\big) = \mathcal{W}_1\big(f_\sharp\big(\phi_\sharp(\mu)\big), f_\sharp\big(\phi_\sharp(\nu)\big)\big)
$$

$$
\begin{aligned}
&= \sup_{g \in L_1(\mathcal{Z}_{\phi'})} \int_{\mathcal{Z}_{\phi'}} g(z) \, \mathrm{d}\big(f_\sharp\big(\lambda_\sharp(\mu)\big) - f_\sharp\big(\lambda_\sharp(\nu)\big)\big)(z) \\
&= \sup_{g \in L_1(\mathcal{Z}_{\phi'})} \int_{\mathcal{Z}_{\phi'}} g(z) \, \mathrm{d}f_\sharp(\phi_\sharp(\mu) - \phi_\sharp(\nu))(z) \\
&= \sup_{g \in L_1(\mathcal{Z}_{\phi'})} \int_{\mathcal{Z}_{\phi'}} g \circ f(z) \, \mathrm{d}(\phi_\sharp(\mu) - \phi_\sharp(\nu))(z) \\
&= \mathsf{Lip}(f) \sup_{g \in L_1(\mathcal{Z}_{\phi'})} \int_{\mathcal{Z}_{\phi'}} \frac{g \circ f(z)}{\mathsf{Lip}(f)} \mathrm{d}(\mu - \nu)(z) \\
&\leq \mathsf{Lip}(f) \sup_{h \in L_1(\mathcal{Z}_{\phi})} \int_{\mathcal{Z}_{\phi}} h(x) \, \mathrm{d}(\mu - \nu)(z) \\
&= \mathsf{Lip}(f) \cdot \mathcal{W}_1\big(\phi_\sharp(\mu), \phi_\sharp(\nu)\big),
\end{aligned}
$$

as desired. $\qquad\square$

## D   PROOFS AND DETAILS OF SECTION 5

We start by restating Theorem 2 from Chuang et al. (2021) using encoders $\phi$ from some general set $\mathcal{X}$ to $\mathcal{Z}$.

**Theorem D.1** (Theorem 2 in Chuang et al. (2021)). *Fix $\gamma > 0$ and an encoder $\phi : \mathcal{X} \to \mathcal{Z}$. Then, for every distribution $\mu$ on $\mathcal{X} \times \mathcal{Y}$, for every predictor $\psi = (\psi_y)_{i \in \mathcal{Y}}$ and every $\delta \in (0, 1)$, with probability at least $1 - \delta$ over all choices of $\mathcal{S} \sim \mu^m$, we have that the generalization gap $R_\mu(\psi \circ \phi) - \hat{R}_{\gamma, \mathcal{S}}(\psi \circ \phi)$ is upper bounded by*

$$
\mathbb{E}_{c \sim \mu_y} \left[ \frac{\mathsf{Lip}\left(\rho_\psi(\cdot, c)\right)}{\gamma} \mathbb{E}_{T, \tilde{T} \sim \mu_c^{m_c}} \left[ \mathcal{W}_1\big(\phi_\sharp(\mu_{c,T}), \phi_\sharp(\mu_{c,\tilde{T}})\big) \right] \right] + \sqrt{\frac{\log(1/\delta)}{2m}},
$$

*where for each $c \in \mathcal{Y}$, $m_c$ denotes the number of pairs $(X, c)$ in $\mathcal{S}$. Also, recall that for $T \sim \mu_c^{m_c}$, $\mu_{c,T}$ is the empirical distribution $\mu_{c,T} := \sum_{X \in T} \delta_X$; similarly for $\mu_{c,\tilde{T}}$.*

To obtain Theorem 5.1 we replace $\mathcal{X}$ by $\mathcal{G}$ and consider graph encoders $\phi : \mathcal{G} \to \mathcal{Z}_\phi$ and $\lambda : \mathcal{G} \to \mathcal{Z}_\lambda$ such that $\lambda$ upper bounds $\phi$ in expressive power. Then, Lemma 4.2 ensures the existence of $f$ such that $\phi = f \circ \lambda$ and Proposition 4.4 consequently implies $\mathcal{W}_1\big(\phi'_\sharp(\mu_{c,T}), \phi'_\sharp(\mu_{c,\tilde{T}})\big) \leq \mathsf{Lip}(f) \cdot \mathcal{W}_1\big(\phi_\sharp(\mu_{c,T}), \phi_\sharp(\mu_{c,\tilde{T}})\big)$ for any $T, \tilde{T} \sim \mu_c^{m_c}$. Plugging this into the bound above results in the bound given in Theorem 5.1.

While the bound in Theorem 5.1 is theoretically useful, the expectation term over $T, \tilde{T} \sim \mu_c^{m_c}$ is intractable in general. To address this drawback, we derive another bound in Theorem D.2, which can be computed via sampling in practice and is the one used in our experiments.

**Theorem D.2.** *Let $\{T^j, \tilde{T}^j\}_{j=1}^n$ be $n$ pairs of graph samples where each $T^j, \tilde{T}^j \sim \mu_c^{\lfloor m_c/2n \rfloor}$, $m = \sum_{c=1}^K \lfloor m_c/2n \rfloor$, and $\Delta(\cdot)$ be the diameter of a space. For any Lipschitz continuous function $f : \mathcal{Z}_\phi \to \mathcal{Z}_\lambda$ such that $\phi = f \circ \lambda$, with probability at least $1 - \delta$ for samples $\mathcal{S} \sim \mu^m$, we have*

$$
R_\mu(\psi \circ \phi) - \hat{R}_{\gamma, \mathcal{S}}(\psi \circ \phi) \leq \sqrt{\frac{\log(2/\delta)}{2m}} +
$$
$$
\mathbb{E}_{c \sim \mu_y} \left[ \frac{\mathsf{Lip}\left(\rho_\psi(\cdot, c)\right) \mathsf{Lip}(f)}{\gamma} \left( \frac{1}{n} \sum_{j=1}^n \mathcal{W}_1\big(\lambda_\sharp(\mu_{c,T^j}), \lambda_\sharp(\mu_{c,\tilde{T}^j})\big) + 2\Delta(\lambda_\sharp(\mu_c)) \sqrt{\frac{\log(2K/\delta)}{n \lfloor m_c/2n \rfloor}} \right) \right].
$$

The proof is again a consequence of Lemma 4.2 and Proposition 4.4, but this time relying on Corollary 6 in Chuang et al. (2021). We note that the diameter will be bounded when $B$-bounded graph encoders are considered.

We conclude with the proof of Proposition 5.2.

---

**Algorithm 1:** An algorithm to compute the bound in Theorem D.2

> **Input** : $\delta$, $m_c$, $n$, $\gamma$, $K$, $\mathcal{S}$, $\lambda$, $\phi$ and $L_\rho$ (Lipschitz constant of $\rho_\psi(;c)$)
> **Output** : Bound
> 1 $L_f \leftarrow 0$;
>    // *Estimate* $\mathsf{Lip}(f)$
> 2 **for** *all* $G, H \in \mathcal{S}$ *and* $\lambda(G) \neq \lambda(H)$ **do**
> 3     $r \leftarrow \frac{\|\phi(G) - \phi(H)\|}{\|\lambda(G) - \lambda(H)\|}$;
> 4     $L_f \leftarrow \max(r, L_f)$;
> 5 **end**
> 6 $b \leftarrow 0$;
> 7 **for** $c \leftarrow 1, \dots, K$ **do**
> 8     $w_c \leftarrow 0$;
> 9     **for** $j \leftarrow 1, \dots, n$ **do**
> 10        Randomly sample $\{G_i\}_{i=1}^{2m_c}$ from graphs of the class $c$ in $\mathcal{S}$;
>          // *Compute 1-Wasserstein using the Hungarian method*
> 11        $w_c \leftarrow w_c + \mathcal{W}_1\big(\{\lambda(G_i)\}_{i=1}^{m_c}, \{\lambda(G_i)\}_{i=m_c+1}^{2m_c}\big)$;
> 12     **end**
> 13     $w_c \leftarrow w_c / n$;
> 14     $\Delta_c \leftarrow 0$;
>       // *Estimate* $\Delta(\lambda_\sharp(\mu_c))$
> 15     **for** *all* $G, H \in \mathcal{S}$ *and* $G, H$ *belong to the class* $c$ **do**
> 16        $\Delta_c = \max(\Delta_c, \|\lambda(G) - \lambda(H)\|)$;
> 17     **end**
> 18     $b = b + \frac{L_\rho L_f}{\gamma}\big(w_c + 2\Delta_c \sqrt{\frac{\log(2K/\delta)}{n\lfloor m_c/2n \rfloor}}\big)$;
> 19 **end**
> 20 $m = K\lfloor m_c/2n \rfloor$;
> 21 **return** $\frac{b}{K} + \sqrt{\frac{\log(2/\delta)}{2m}}$;

---

**Proposition 5.2.** *Under the same assumptions as in Theorem 5.1, but with the additional requirement that the predictors $\psi_c$ in $\psi$ are Lipschitz, and that the bounding graph classifier $\lambda$ has a large margin, i.e., $\rho_\psi(\lambda(G), y) \geq \gamma$ for all $(G, y) \sim \mu$, then for any $\delta \in (0, 1)$, with probability at least $1 - \delta$ over all choices $\mathcal{S} \sim \mu^m$, we have that the generalization bound given in Theorem 5.1 is lower bound by*

$$\frac{\mathsf{Lip}(f) \cdot \mathbb{E}_{c \sim \mu_y}\left[\mathsf{Lip}(\rho_\psi(\cdot, c))\mathbb{E}_{T, \tilde{T} \sim \mu_c^{m_c}}\left[\mathcal{W}_1\left(\lambda_\sharp(\mu_{c,T}), \lambda_\sharp(\mu_{c,\tilde{T}})\right)\right]\right]}{(\min_{c \in \mathcal{Y}} \mathsf{Lip}(\psi_c))(\min_{c,c' \in \mathcal{Y}, c \neq c'} \mathcal{W}_1(\lambda_\sharp(\mu_c), \lambda_\sharp(\mu_{c'})))} + \sqrt{\frac{\log(1/\delta)}{2m}}.$$

*Proof.* Since we assume the margin $\gamma$ is satisfied for $\psi \circ \lambda$, for all graph samples, and for each $c \in \mathcal{Y}$, the predictor $\psi_c \in \psi$ is Lipschitz, then (see Lemma 10 in Chuang et al. (2021)) we have

$$\gamma \leq \big(\min_{\substack{c,c' \in \mathcal{Y} \\ c \neq c'}} \mathcal{W}_1(\lambda_\sharp(\mu_c), \lambda_\sharp(\mu_{c'}))\big)\big(\min_{c \in \mathcal{Y}} \mathsf{Lip}(\psi_c)\big).$$

In other words,

$$\frac{1}{\big(\min_{\substack{c,c' \in \mathcal{Y} \\ c \neq c'}} \mathcal{W}_1(\lambda_\sharp(\mu_c), \lambda_\sharp(\mu_{c'}))\big)\big(\min_{c \in \mathcal{Y}} \mathsf{Lip}(\psi_c)\big)} \leq \frac{1}{\gamma}. \qquad (*)$$

Furthermore, we know from Theorem 5.1 that for every $\delta \in (0, 1)$, with probability at least $1 - \delta$ over all choices of $\mathcal{S} \sim \mu^m$, we have that the generalization gap $R_\mu(\psi \circ \phi) - \hat{R}_{\gamma, \mathcal{S}}(\psi \circ \phi)$ is upper bounded by

$$\mathbb{E}_{c \sim \mu_y}\left[\frac{\mathsf{Lip}(\rho_\psi(\cdot, c))\mathsf{Lip}(f)}{\gamma}\mathbb{E}_{T, \tilde{T} \sim \mu_c^{m_c}}\left[\mathcal{W}_1\big(\lambda_\sharp(\mu_{c,T}), \lambda_\sharp(\mu_{c,\tilde{T}})\big)\right]\right] + \sqrt{\frac{\log(1/\delta)}{2m}}.$$

Since $\mathsf{Lip}(f)$ and $\big(\min_{\substack{c,c'\in\mathcal{Y}\\c\neq c'}}\mathcal{W}_1(\lambda_\sharp(\mu_c),\lambda_\sharp(\mu_{c'}))\big)\big(\min_{c\in\mathcal{Y}}\mathsf{Lip}(\psi_c)\big)$ are independent of $c\sim\mu_y$, we can take them out of the expectation, that is we rewrite the upper bound as

$$\frac{\mathsf{Lip}(f)}{\gamma}\mathbb{E}_{c\sim\mu_y}\left[\mathsf{Lip}\left(\rho_\psi(\cdot,c)\right)\mathbb{E}_{T,\tilde{T}\sim\mu_c^{m_c}}\left[\mathcal{W}_1\big(\lambda_\sharp(\mu_{c,T}),\lambda_\sharp(\mu_{c,\tilde{T}})\big)\right]\right]+\sqrt{\frac{\log(1/\delta)}{2m}}.$$

Finally, by replacing $\frac{1}{\gamma}$ with the lower bound $(*)$ we get that the generalization upper bound is lower bounded by

$$\frac{\mathsf{Lip}(f)\cdot\mathbb{E}_{c\sim\mu_y}\left[\mathsf{Lip}(\rho_\psi(\cdot,c))\mathbb{E}_{T,\tilde{T}\sim\mu_c^{m_c}}\left[\mathcal{W}_1\left(\lambda_\sharp(\mu_{c,T}),\lambda_\sharp(\mu_{c,\tilde{T}})\right)\right]\right]}{(\min_{c\in\mathcal{Y}}\mathsf{Lip}(\psi_c))(\min_{c,c'\in\mathcal{Y},c\neq c'}\mathcal{W}_1(\lambda_\sharp(\mu_c),\lambda_\sharp(\mu_{c'})))}+\sqrt{\frac{\log(1/\delta)}{2m}},$$

as desired. □

## E    COMPUTATION OF GENERALIZATION BOUNDS

**VC based bound (Morris et al., 2023):** We use the classical bounds on the generalisation gap based on the VC-dimension Vapnik and Chervonenkis (1964); Vapnik (1998). That is, with probability $1-\delta$, the generalisation gap is bounded by

$$\sqrt{\frac{1}{|\mathcal{S}|}\Big(D\big(\log(2|\mathcal{S}|/D)+1\big)-\log(\delta/4)\Big)}$$

where $|\mathcal{S}|$ is the sample size and $D$ is the VC dimension. In our setting, results by Morris et al. (2023) implies that $D$ is bounded by the number of graphs, distinguishable by the hypothesis class. In our experiments, we computed the latter as the number of graphs in $\mathcal{S}$ distinguishable by 1-WL at each iteration.

**PAC-Bayesian bound (Ju et al., 2023)** We follow Ju et al. (2023) to compute the bound, that is, with probability $1-\delta$, the generalisation gap is bounded by

$$\sum_{\ell=l}^{L}\sqrt{\frac{CB_{\text{loss}}d_\ell(\max_{G\sim\mu}||\mathbf{X}_G||^2||\mathbf{A}_G||^{2(l-1)}(r_\ell^2\prod_{j=1}^{L}s_j^2)}{|\mathcal{S}|}}+O(\frac{\log(\delta^{-1})}{|\mathcal{S}|^{3/4}}),$$

where $L$ is the number of MPNN layers, $B_{\text{loss}}$ is a cap on the value of the loss function, $C$ is a fixed Lipschitz constant depending on the activation and loss functions, and $|\mathcal{S}|$ is again the sample size. Moreover, $d_\ell$ is the second dimension of the weight matrix $W^{(\ell)}$ at layer $\ell$, $\mathbf{X}_G$ is the vertex feature matrix of $G$ and $\mathbf{A}_G$ is the adjacency matrix of $G$. For MPNNs, $s_j=1$, $r_\ell=||\mathbf{W}^{(\ell)}||_F$ where $||\mathbf{W}^{(\ell)}||_F$ is the Frobenius norm of $\mathbf{W}^{(\ell)}$.

**Our bound** In practice, we estimate $\mathsf{Lip}(f)$ and $\Delta(\lambda_\sharp(\mu_c))$ in Theorem D.2 using data in the training sets, thus both can be computed in $O(|\mathcal{S}|^2)$. The 1-Wasserstein distance can be computed in $O((\frac{m_c}{2n})^3)$ using the Hungarian method (Kuhn, 1955). Normally we have $|\mathcal{S}|^2\ll(\frac{m_c}{2n})^3$ because $|S|=K\lfloor m_c/2n\rfloor$ and $\frac{m_c}{2n}>1$. So the total time complexity to compute the bound is $O((\frac{m_c}{2n})^3)$ which is tractable for most datasets. For very large datasets, practitioners can choose to use a smaller $m_c$ and a larger $n$ to reduce the computational cost. An algorithm to compute the bound is sketched in Algorithm 1.

## F    ADDITIONAL EXPERIMENTAL RESULTS

The results of graph classification with embedding normalization are shown in Table 3. The empirical loss gaps are plotted alongside our bounds in Figure 2. In Table 4, we list the correlation coefficients between the empirical loss gaps and the three generalization bounds for different $\mathcal{F}$-MPNN variants. Our bounds consistently show a positive correlation with the empirical loss gaps, while the VC and PAC bounds show a negative correlation in most cases. The code implementation is available at https://github.com/seanli3/hom_gen.

Table 3: Graph classification gaps with different numbers of MPNN layers. The MPNN embeddings are normalized.

| # Layers | | ENZYMES | PROTEINS | Dataset MUTAG | SIDER | BACE |
|---|---|---|---|---|---|---|
| 1 | Loss gap | $0.105_{\pm0.010}$ | $-0.018_{\pm0.009}$ | $-0.091_{\pm0.017}$ | $0.013_{\pm0.013}$ | $-0.004_{\pm0.010}$ |
| | Our bound | $0.800_{\pm0.095}$ | $2.203_{\pm0.134}$ | $1.101_{\pm0.063}$ | $1.137_{\pm0.552}$ | $1.147_{\pm0.143}$ |
| | VC dimension | 586 | 929 | 51 | 960 | 621 |
| | VC bound | $1.302_{\pm0.000}$ | $1.292_{\pm0.001}$ | $1.100_{\pm0.004}$ | $1.302_{\pm0.000}$ | $1.301_{\pm0.000}$ |
| | PAC bound | $3.48_{\pm0.01}$ | $5.04_{\pm0.00}$ | $3.06_{\pm0.05}$ | $52.39_{\pm1.86}$ | $21.525_{\pm1.072}$ |
| 2 | Loss gap | $0.098_{\pm0.022}$ | $-0.023_{\pm0.011}$ | $-0.097_{\pm0.019}$ | $0.015_{\pm0.006}$ | $0.000_{\pm0.010}$ |
| | Our bound | $0.586_{\pm0.036}$ | $1.016_{\pm0.035}$ | $1.208_{\pm0.046}$ | $1.017_{\pm0.644}$ | $1.089_{\pm0.135}$ |
| | VC dimension | 595 | 996 | 121 | 1300 | 1060 |
| | VC bound | $1.302_{\pm0.000}$ | $1.292_{\pm0.000}$ | $1.281_{\pm0.003}$ | $1.302_{\pm0.000}$ | $1.302_{\pm0.000}$ |
| | PAC bound | $12.75_{\pm0.22}$ | $31.94_{\pm2.79}$ | $8.17_{\pm0.12}$ | $132.79_{\pm8.12}$ | $51.573_{\pm2.853}$ |
| 3 | Loss gap | $0.118_{\pm0.023}$ | $-0.027_{\pm0.011}$ | $-0.083_{\pm0.006}$ | $0.030_{\pm0.008}$ | $-0.006_{\pm0.015}$ |
| | Our bound | $0.572_{\pm0.024}$ | $0.834_{\pm0.015}$ | $0.993_{\pm0.039}$ | $1.221_{\pm0.957}$ | $1.167_{\pm0.610}$ |
| | VC dimension | 595 | 996 | 135 | 1309 | 1089 |
| | VC bound | $1.302_{\pm0.000}$ | $1.293_{\pm0.000}$ | $1.286_{\pm0.002}$ | $1.302_{\pm0.000}$ | $1.302_{\pm0.000}$ |
| | PAC bound | $56.98_{\pm1.06}$ | $276.78_{\pm0.00}$ | $21.96_{\pm0.00}$ | $341.04_{\pm19.89}$ | $124.605_{\pm7.506}$ |
| 4 | Loss gap | $0.129_{\pm0.007}$ | $-0.004_{\pm0.005}$ | $-0.087_{\pm0.011}$ | $0.026_{\pm0.015}$ | $0.001_{\pm0.024}$ |
| | Our bound | $0.573_{\pm0.027}$ | $0.847_{\pm0.027}$ | $0.848_{\pm0.085}$ | $1.039_{\pm0.898}$ | $0.705_{\pm0.026}$ |
| | VC dimension | 595 | 996 | 139 | 1309 | 1093 |
| | VC bound | $1.302_{\pm0.000}$ | $1.292_{\pm0.001}$ | $1.291_{\pm0.002}$ | $1.302_{\pm0.000}$ | $1.302_{\pm0.000}$ |
| | PAC bound | $308.43_{\pm0.00}$ | $2331.63_{\pm0.00}$ | $57.69_{\pm1.83}$ | $845.62_{\pm73.11}$ | $310.732_{\pm12.520}$ |
| 5 | Loss gap | $0.169_{\pm0.014}$ | $0.003_{\pm0.035}$ | $-0.086_{\pm0.013}$ | $0.006_{\pm0.038}$ | $0.002_{\pm0.014}$ |
| | Our bound | $0.575_{\pm0.039}$ | $0.713_{\pm0.246}$ | $0.799_{\pm0.051}$ | $0.923_{\pm0.438}$ | $0.703_{\pm0.012}$ |
| | VC dimension | 595 | 996 | 139 | 1309 | 1093 |
| | VC bound | $1.302_{\pm0.000}$ | $1.292_{\pm0.001}$ | $1.292_{\pm0.002}$ | $1.302_{\pm0.000}$ | $1.302_{\pm0.000}$ |
| | PAC bound | $1615.10_{\pm89.11}$ | $17992.81_{\pm4950.10}$ | $155.74_{\pm4.68}$ | $2179.21_{\pm190.74}$ | $744.08_{\pm31.12}$ |
| 6 | Loss gap | $0.169_{\pm0.023}$ | $-0.002_{\pm0.032}$ | $-0.104_{\pm0.008}$ | $0.029_{\pm0.009}$ | $-0.013_{\pm0.015}$ |
| | Our bound | $0.603_{\pm0.032}$ | $0.793_{\pm0.136}$ | $0.778_{\pm0.049}$ | $1.192_{\pm0.561}$ | $0.679_{\pm0.018}$ |
| | VC dimension | $1.302_{\pm0.000}$ | $1.292_{\pm0.001}$ | $1.291_{\pm0.002}$ | $1.302_{\pm0.000}$ | $1.302_{\pm0.000}$ |
| | VC bound | $1.302_{\pm0.000}$ | $1.292_{\pm0.001}$ | $1.291_{\pm0.002}$ | $1.302_{\pm0.000}$ | $1.302_{\pm5.870}$ |
| | PAC bound | $8931.00_{\pm0.00}$ | $135762.52_{\pm59439.71}$ | $410.31_{\pm17.44}$ | $5254.88_{\pm655.89}$ | $1860.94_{\pm5.96}$ |

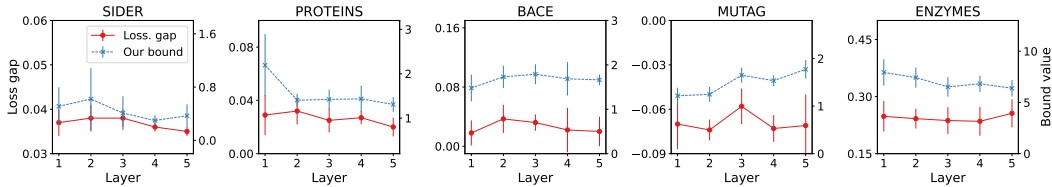

Figure 2: Loss gaps and bounds of MPNNs of different layers

Table 4: Correlation coefficients between the empirical loss gaps and three generalization bounds on difference $\mathcal{F}$-MPNN variants.

| Dataset | | $\{C_3\}$-MPNN | $\{C_4\}$-MPNN | $\{C_5\}$-MPNN | $\mathcal{F}$-MPNN variants $\{C_6\}$-MPNN | $\{K_4\}$-MPNN | $\{K_5\}$-MPNN | $\{K_6\}$-MPNN |
|---|---|---|---|---|---|---|---|---|
| SIDER | VC bound | 0.347 | -0.352 | -0.388 | -0.327 | -0.480 | -0.164 | -0.731 |
| | PAC bound | -0.421 | -0.889 | -0.770 | -0.623 | -0.904 | -0.573 | -0.565 |
| | Our bound | 0.545 | 0.414 | 0.464 | 0.537 | 0.897 | 0.539 | 0.925 |
| BACE | VC bound | -0.469 | -0.251 | -0.292 | -0.448 | -0.251 | -0.251 | -0.251 |
| | PAC bound | -0.545 | 0.589 | -0.955 | 0.118 | -0.340 | -0.090 | 0.397 |
| | Our bound | 0.902 | 0.104 | 0.303 | 0.238 | 0.340 | 0.221 | 0.502 |

