# OpenReview forum: "Towards Bridging Generalization and Expressivity of Graph Neural Networks"
_ICLR.cc/2025/Conference — ICLR 2025 Poster_

### Official Review · Reviewer_UQK4 · 2024-10-27

**Soundness:** 2
**Presentation:** 2
**Contribution:** 3
**Rating:** 5
**Confidence:** 3

**Summary:**

This paper theoretically studies the relationship between the generalization of GNN and the expressive power of GNN in terms of the variance of graph structures. This work especially shows a trade-off between intra-class concentration and inter-class separation, which is the key insight into the generalization gap. Some experiments are conducted to support the theory.

**Strengths:**

1. Some theoretical results seem solid and strong.
2. The problem to study is very interesting and important to the community to bridge the gap between the generalization and expressive power of GNN.

**Weaknesses:**

1. The theoretical proof is not complete. The proof of Corollary 4.5 and Proposition 5.2 are missing.
2. The paper is full of equations. Section 3 is not easy to follow when introducing so many terminologies with any examples.
3. The practical insights are limited. Section 6 is quite confusing. This section shows that different graphs can lead to opposite effects on inter-class separation by the expressive power. Then, what should we do in practice based on this knowledge? Moreover, Section 6 seems irrelevant to intra-class concentration. Then, is the case study of Section 6 meaningful enough?
4. Results in Table 2 look weak. I cannot see that the proposed bound is better than VC bound in simulating the trend of accuracy or loss gap.

Minor: 1. at the end of line 482, it shall be "Figure 2(b)" instead of "Figure 2(a)"?
2. Please resize Figure 2 to make it look nicer.

**Questions:**

In Figure 2 (a), why not show both the loss gap and the bound for each dataset? In Figure 2 (b), why not show both $W_1$ and $Lip(f)$ for each dataset?

---

> ### Author Response · Authors · 2024-11-24
>
> > S1 Some theoretical results seem solid and strong. The problem to study is very interesting and important to the community to bridge the gap between the generalization and expressive power of GNN.
>
> We thank the reviewer for the positive comments and appreciate that the reviewers finds the results strong and the problem interesting.

---

> ### Author Response · Authors · 2024-11-24
> **Proof details.**
>
> > W1. The theoretical proof is not complete. The proof of Corollary 4.5 and Proposition 5.2 are missing.
>
> We revised the text so that the proofs of Corollary 4.5 and Proposition 5.2 are now more clearly explained. That is, for Corollary 4.5 we now write “Indeed, Proposition 4.3 implies $\text{Lip}(f)=B/S$. Combined with Proposition 4.4., this gives $\mathcal W_1(\phi_\sharp'(\nu), \phi_\sharp'(\nu'))\leq \text{Lip}(f) \cdot \mathcal W_1(\phi_\sharp(\nu), \phi_\sharp(\nu'))= (B/S)\cdot \mathcal W_1(\phi_\sharp(\nu), \phi_\sharp(\nu'))$“ instead of simply saying it is a consequence of the Proposition 4.3 and 4.4.
>
> Similarly, for Proposition 5.2., instead of simply saying that the lower bound for $1/\gamma$ and equation $(\dagger)$ results in that proposition, we say “By replacing $1/\gamma$ in $(\dagger)$ by this bound, we obtainProposition 5.2, see Appendix D for details” and also have detailed the proof in appendix D. The changes are highlighted in the revision.

---

> ### Author Response · Authors · 2024-11-24
> **Examples**
>
> > W2 The paper is full of equations. Section 3 is not easy to follow when introducing so many terminologies with any examples.
>
> Thanks for your suggestion. We have added examples in Appendix B for the homomorphism and 1-Wasserstein distance, hopefully it makes the concepts more clear.

---

> ### Author Response · Authors · 2024-11-24
> **Practical insights**
>
> > W3 The practical insights are limited. Section 6 is quite confusing. This section shows that different graphs can lead to opposite effects on inter-class separation by the expressive power. Then, what should we do in practice based on this knowledge? Moreover, Section 6 seems irrelevant to intra-class concentration. Then, is the case study of Section 6 meaningful enough?
>
> Thank you for your comments. Our work makes two key practical contributions:
> - We propose a generalization bound that empirically quantifies the generalization ability of graph encoders (Theorem 5.1, particularly the sample-based version in Theorem D.2, Appendix D).
> - The lower bound in Proposition 5.2 identifies two main factors influencing generalization: intra-class concentration and inter-class separation.
>
> The purpose of Section 6 is to illustrate how these factors, in relation to graph encoder expressivity, affect generalization.
>
> - **Intra-class concentration:** In Table 1, we analyze the intra-class concentration for a class $c$, where the graphs in this class are assumed to cluster around two representative graphs $G$ and $G'$. The intra-class concentration is estimated using the Wasserstein distance $\mathcal W_1(\lambda_\sharp(G), \lambda_\sharp(G'))$, shown in the last column of Table 1. The results reveal that, for the same graphs, increasing the expressive power of graph encoders can have opposite effects on intra-class concentration, demonstrating the nuanced behavior of expressive models.
> - **Inter-class separation:** The final paragraph of Section 6 addresses inter-class separation, measured by the Wasserstein distance $\max_{c, c' \in \mathcal Y, c \neq c'} \mathcal W_1(\lambda_\sharp(\mu_c), \lambda_\sharp(\mu_{c'}))$. We show that inter-class separation can influence the generalization bound in two opposite directions, depending on the distribution of class representations.
>
> Both factors—**intra-class concentration** and **inter-class separation**—are thus central to Section 6. We would appreciate further clarification on the reviewer’s comment that “Section 6 seems irrelevant to intra-class concentration” to respond more effectively.
>
> Finally, our results also provide insights into hyperparameter selection. For instance, we analyze how graph homomorphism patterns or layers influence generalization. The generalization ability of a GNN can be evaluated through its graph encoder (e.g., 1-WL, k-WL, F-WL) without training the model, enabling efficient and explainable hyperparameter selection. Indeed, such discrete graph encoders are fast to run and easy to analyze. This insight is illustrated in Figure 1, where different choices of graph patterns and layers lead to bounds that closely reflect actual generalization gaps. This approach offers practical guidance for model design and hyperparameter tuning.

---

> ### Author Response · Authors · 2024-11-24
> **Results**
>
> > W4 Results in Table 2 look weak. I cannot see that the proposed bound is better than VC bound in simulating the trend of accuracy or loss gap.
>
> The generalization bound based on the VC-dimension depends monotonically on the VC dimension (see Appendix E). However, the VC dimension itself is monotonically non-decreasing with respect to the number of layers and becomes constant after a certain depth. This implies that the VC-dimension does not effectively capture a model's generalization ability, as deeper models often generalize well, as demonstrated in Table 2.
> In contrast, our proposed bound is non-monotonic and provides a better indication of trends in generalization gaps. While these trends may not be immediately evident from the raw numbers in Table 2, we address this by adding correlation matrices between the bounds and empirical gaps alongside the table. These matrices highlight that our bounds correlate much more strongly with empirical gaps compared to other bounds.
> Additionally, in Figure 2 of Appendix G, we plot our bound against the loss gap, providing a visual demonstration of the strong correlation between our bounds and empirical gaps. Notably, our bound shows weaker correlation on the ENZYMES dataset. This is due to ENZYMES being a small dataset containing only 600 graphs across 5 classes, resulting in relatively small sample sizes per class ($m_c$ in Theorem D.2). This leads to greater uncertainty in bounding the generalization gap, manifesting in larger bound values and higher variance.

---

> ### Author Response · Authors · 2024-11-24
> **Minor**
>
> > Minor: 1. at the end of line 482, it shall be "Figure 2(b)" instead of "Figure 2(a)"? 2. Please resize Figure 2 to make it look nicer.
>
> Thank you for the suggestion and pointing out this typo. We resized Figure 2, as requested. We also  combined Figure 1 with Figure 2 to save some space.
>
> > Minor 2. In Figure 2 (a), why not show both the loss gap and the bound for each dataset? In Figure 2 (b), why not show both W1 and Lip(f)  for each dataset?
>
> In the original submission, we presented both the loss gap (bars) and bounds (lines) in Figure 2(a), as well as W1 (bars) and Lip(f) (lines) in Figure 2(b). We understand the reviewer’s confusion, particularly because Figure 1 used an all-bar representation. In the revision, we have unified the representation to consistently use bars and provided a clearer explanation of the plot legends. Additionally, we combined Figures 1 and 2 to save space and improve clarity. We appreciate the reviewer for highlighting this issue.

---

> ### Author Response · Authors · 2024-11-27
>
> Whenever you have a chance, we’d appreciate your feedback on our rebuttal - thank you.

---

> > ### Comment · Reviewer_UQK4 · 2024-11-28
> >
> > Thank you for the response. I have further questions if my understanding is correct.
> >
> > 1. Weakness 3: Now I get that Section 6 is related to intra-class concentration. The authors lead me to reading Figure 1, but I cannot follow. What are the differences between the figures in the first row? In the discussion, the authors mention the results for ENZYMES and PROTEINS. Which figures in Figure 1 correspond to these two datasets? The caption does not mention these two datasets.
> >
> > 2. Weakness 4: I don't think Figure 2 shows a similar trend between your derived bound and the generalization gap. There is no clear trend in the generalization gap shown in Figure 2 since the generalization gap differences between different layers are small in these datasets. At least for PROTEINS, SIDER, and MUTAG, I think the trends of the generalization gap are constant. Then, VC bound is also constant in these three cases and is overall smaller than your bound in PROTEINS and MUTAG.

---

> ### Author Response · Authors · 2024-11-28
>
> Thank you very much for reply and your two follow-up questions, which we now answer in turn.
>
> > Weakness 3: Now I get that Section 6 is related to intra-class concentration. The authors lead me to reading Figure 1, but I cannot follow. What are the differences between the figures in the first row? In the discussion, the authors mention the results for ENZYMES and PROTEINS. Which figures in Figure 1 correspond to these two datasets? The caption does not mention these two datasets.
>
> Thanks for pointing out our oversight. The figures’ titles were left out due to our mistake. For that figure: Top row from left to right: PROTEINS 4-layer MPNN, PROTEINS 6-layer MPNN, ENZYMES 4-layer MPNN, ENZYMES 6-layer MPNN. Thanks for catching this. We will update the figure and caption accordingly. (The deadline for submitting a revised version has passed so we cannot upload our changed revision at this point in time.)
>
> > Weakness 4: I don't think Figure 2 shows a similar trend between your derived bound and the generalization gap. There is no clear trend in the generalization gap shown in Figure 2 since the generalization gap differences between different layers are small in these datasets. At least for PROTEINS, SIDER, and MUTAG, I think the trends of the generalization gap are constant. Then, VC bound is also constant in these three cases and is overall smaller than your bound in PROTEINS and MUTAG.
>
> Thank you for your comment. As the reviewer rightly observes, the generalization gaps in our experiments only fluctuate slightly: For SIDER, the gap decreases from 0.037 to 0.034; For ENZYMES, it initially decreases from 0.248 at layer 1 to 0.235 at layer 4, then increases to 0.264 at layer 6; For PROTEINS and BACE, it fluctuates within a small range of 0.03 to 0.02. While these changes in loss appear minor, we want to emphasize that  they correspond to **significant variations** in test accuracy, reaching up to 6 percentage points.
>
> Regarding the “trend,” the key point we aim to highlight in Figure 2 is that our bound does not necessarily increase with the number of layers. This demonstrates that our approach is **more adaptable** to the intrinsic properties of the underlying graphs. In contrast, the VC bound is computed using an upper bound on the VC dimension, as described in [1]. This upper bound represents the number of distinguishable graphs by MPNNs with L layers. Notably, the VC bound is non-decreasing with L and becomes constant for sufficiently large L. This is for example nicely illustrated by the BACE results.
>
> For the comparison with the VC bound, it is crucial to clarify that we use an (overly) **optimistic approximation** of the VC dimension. Specifically, as detailed in Appendix E and mentioned earlier, the VC bound is defined in terms of the VC dimension. Theoretically, the VC dimension for L-layer MPNNs is determined by the number of distinguishable graphs of a given size, a number that is exceedingly large and computationally infeasible to determine. Moreover, for the VC-based generalization bound to apply, the sample size must exceed the VC dimension, which would require unrealistically large datasets in our setting. Instead, we approximate the VC dimension by considering the number of distinguishable graphs within the training set. Consequently, our estimated VC dimension is smaller than the sample size, leading to a lower generalization bound than what would result from using the actual VC dimension.
>
> In fact, the computational complexity of estimating the VC dimension underscores a key advantage of our bound: it is more straightforward to compute while remaining effective.
>
> [1] Morris, Geerts,Tönshoff, Grohe. WL meet VC. ICML 2023.
>
> We hope that these answers clarify things a bit more.

---

> > ### Author Response · Authors · 2024-12-01
> > **Further clarification?**
> >
> > As authors can only respond until December 3 by ICLR, we kindly ask the reviewer to let us know if any further clarification is needed. Thanks.

---

> > > ### Comment · Reviewer_UQK4 · 2024-12-01
> > >
> > > I think the emphasized contribution that the derived bound does not necessarily increase with the number of layers is weak. It does not mean a strong result, i.e., the bound provably matches the trend of the loss gap with theoretical guarantee. The empirical results are also not supportive enough since the trend of the loss gap is not clear enough in many datasets. Due to the above reason, I prefer to keep my score of 5.

---

> > > > ### Author Response · Authors · 2024-12-02
> > > >
> > > > Thank you for taking the time to evaluate our responses.
> > > >
> > > > We would like to point out that the correlation matrices in Table 2 clearly show that our results exhibit a stronger correlation between empirical gaps and bounds compared to other bounds across all datasets, except for ENZYMES. There is a reason why our bound has a weaker correlation on ENZYMES. This is because ENZYMES contains 6 classes but only 600 graphs, resulting in a small $m_c$ that increases the bound.
> > > >
> > > > Our bound does provide a valid upper bound on the loss gap (as detailed in Theorem 5.1) with theoretical guarantees. These guarantees are consistent with those commonly found in classical learning theory.
> > > >
> > > > We would also like to emphasize that, to the best of our knowledge, our bound is unique in its ability to adapt to the distribution of the underlying graph properties through the use of the k-Variance. This adaptation establishes a more nuanced connection between generalization and expressiveness than what has been offered by previous bounds. We believe this novel perspective is a significant step forward in understanding the interplay between these two important aspects.

---

### Official Review · Reviewer_31fD · 2024-10-31

**Soundness:** 3
**Presentation:** 2
**Contribution:** 3
**Rating:** 8
**Confidence:** 3

**Summary:**

The authors in this paper study the connection between expressivity and generalization ability of GNNs in the graph classification task. They first show that the generalization ability of a graph encoder can be upper bounded by that of graph more encoders with higher expressivity . After that, they establish generalization bounds for GNNs from a optimal transfer viewpoint. Concretely, the derived bound contain the Wasserstein distance of embeddings between two random subsets sampled from training nodes within the same class. In this way, the generalization ability of GNNs in the graph classification task can be depicted by the concentration of intra-class embeddings and the separation of inter-class embeddings. Finally, the authors also provide empirical bounds that adopting sampling to computed the Wasserstein distance and verify their theoretic results on real-world datasets. The experimental results generally support their derived bounds.

**Strengths:**

The problem studied in this paper is pivotal in the theory of GNNs, i.e., revealing the connection between expressivity and generalization ability of GNNs. This paper makes a non-trivial progress in this problem, and the derived results are convinced and sound. As for the generalization bounds for GNNs in graph classification task, the derived results extend previous results (Chuang et al. 2021) to graph data, which also provides a new perspective to understand the generalization ability of GNNs. And, the experimental results show that the numerical values of the derived empirical bounds are significant sharper than that of VC bounds.

**Weaknesses:**

- The derived generalization bounds for GNNs contain the Lipschitz constants of graph encoders. These constants could be easily evaluated for shallow models, e.g., GCN with two or three layers, yet it may be difficulty to evaluate for more complex GNNs such as graph transformer [1,2].
- The derived bounds do not take the optimization into consideration and thus could not reflect the impact of optimization algorithms on generalization ability. It has been shown that the training trajectory could also affect the generalization ability of models [3]. Although this observation is not yet verified for GNNs, I believe that improving the generalization bounds by involving the analysis of training dynamics could be a promising direction.


[1] Chen et al., Structure-Aware Transformer for Graph Representation Learning. ICML 2022.

[2] Rampášek et al., Recipe for a General, Powerful, Scalable Graph Transformer. NeurIPS 2022.

[3] Fu et al., Learning Trajectories are Generalization Indicators. NeurIPS 2023.

**Questions:**

Q1: In the experiments you only compare your bounds with VC bounds. It is encouraged to also compare with PAC-Bayesian bounds presented in (Ju et al. 2023).

Q2: Providing some examples to illustrate some definitions could help the readers better understand their meanings, e.g., the definitions of homomorphism and graph invariant in Section 3. You could add some figures to demonstrate these definitions more intuitively.

---

> ### Author Response · Authors · 2024-11-24
> **Interesting and non-trivial**
>
> > S1 The problem studied in this paper is pivotal in the theory of GNNs, i.e., revealing the connection between expressivity and generalization ability of GNNs. This paper makes a non-trivial progress in this problem, and the derived results are convinced and sound.
>
> We thank the reviewer for the positive comments.
>
>
> > S2 As for the generalization bounds for GNNs in the graph classification task, the derived results extend previous results (Chuang et al. 2021) to graph data, which also provides a new perspective to understand the generalization ability of GNNs. The experimental results show that the numerical values of the derived empirical bounds are significantly sharper than those of VC bounds.
>
> We are pleased to see that the reviewer gained a new perspective for understanding generalization based on our work.

---

> ### Author Response · Authors · 2024-11-24
> **Lipschitz and optimization-based**
>
> > W1. The derived generalization bounds for GNNs contain the Lipschitz constants of graph encoders. These constants could be easily evaluated for shallow models, e.g., GCN with two or three layers, yet it may be difficult to evaluate for more complex GNNs such as graph transformers  [1,2].
>
> Thank you for the comments. While our generalization bound involves Lipschitz constants, we want to clarify that they do not directly correspond to the graph encoders.
> - **Decoder Component:** The term  $\text{Lip}(\rho_\psi(\cdot, c))$ represents the Lipschitz constant of the graph decoders. This is typically known, as decoders often rely on standard functions like Softmax, or it can be estimated empirically, as suggested in [4].
> - **Encoder Component:** The term  $\text{Lip}(f)$ , where $f$ maps the graph encoder $\lambda$ to the encoder $\phi$, can also be empirically estimated. Specifically, we compute:
> $$\max_{G,H}\frac{d_{\mathcal Y}(\phi(G),\phi(H))}{d_{\mathcal Y’}(\lambda(G),\lambda(H))},$$
> where the maximum is taken over the sample set. This approach only requires the embedding distance ratio, not the exact Lipschitz constants of the graph encoder.
>
> Regarding graph transformers, their ability to distinguish graphs depends heavily on the structural encodings they use (e.g., subtree extractors, subgraph extractors, WL trees, etc.). Whenever graph transformers (such as in [1,2]) have known expressivity bounds,  our framework can evaluate their generalization, as demonstrated in our analysis. Indeed, we can use the combinatorial graph embeddings determined by their expressiveness analysis.
>
> > W2 The derived bounds do not take the optimization into consideration and thus could not reflect the impact of optimization algorithms on generalization ability. It has been shown that the training trajectory could also affect the generalization ability of models [3]. Although this observation has not yet been verified for GNNs, I believe that improving the generalization bounds by involving the analysis of training dynamics could be a promising direction.
>
>
> The reviewer is absolutely right that trajectory information and the inclusion of (stochastic) gradient descent based optimization may result in a more fine-grained generalization analysis. The provided reference [3] may be an interesting starting point to explore this direction in the graph setting. We note that in the graph setting, Franks et al [5] explore similar ideas, but based on the margin-based analysis by Ji and Telgarsky [6]. However, that analysis works only for deep linear models, without linearities.
>
> [1] Chen et al., Structure-Aware Transformer for Graph Representation Learning. ICML 2022.
>
> [2] Rampášek et al., Recipe for a General, Powerful, Scalable Graph Transformer. NeurIPS 2022.
>
> [3] Fu et al., Learning Trajectories are Generalization Indicators. NeurIPS 2023.
>
> [4] Chuang et al., Measuring Generalization with Optimal Transport. NeurIPS 2021
>
> [5] Franks et al.,  Weisfeiler-Leman at the margin: When more expressivity matters. ICML 2024.
>
> [6] Ji and Telgarsky, Gradient Descent Aligns the Layers of Deep Linear Networks. ICLR 2019.

---

> ### Author Response · Authors · 2024-11-24
> **Questions**
>
> > Q1: In the experiments, you only compare your bounds with VC bounds. It is encouraged to also compare with PAC-Bayesian bounds presented in (Ju et al. 2023).
>
> Thank you for the suggestion. We have now included PAC-Bayesian bounds in our comparison, see Table 2 in the paper, and Table 3 in the appendix. We observe that the PAC bound grows exponentially with layers and shows  a negative correlation with empirical gap. In contrast, our bound better reflects loss gap changes and are positively correlated. In Appendix E, we detail how the PAC-Bayesian bounds are computed based on Ju et al.  2023.
>
> > Q2: Providing some examples to illustrate some definitions could help the readers better understand their meanings, e.g., the definitions of homomorphism and graph invariant in Section 3. You could add some figures to demonstrate these definitions more intuitively.
>
> Thank you for the suggestion. We now provide a new appendix B containing additional examples to demonstrate concepts in Section 3, hopefully it helps to make things clearer.

---

> ### Comment · Reviewer_31fD · 2024-11-27
> **Reply to the Authors**
>
> Thanks the authors for their responses. My previous concerns are adequately addressed. I have increased my score to 8.

---

> > ### Author Response · Authors · 2024-11-27
> >
> > We are glad the reviewer is satisfied with our responses. The improved score is greatly appreciated. Thank you!

---

### Official Review · Reviewer_yEuE · 2024-11-02

**Soundness:** 3
**Presentation:** 3
**Contribution:** 3
**Rating:** 6
**Confidence:** 2

**Summary:**

This paper studies the generalization of graph neural networks. The paper is based on three technical tools. The first one is margin bound, which is common in deriving generalization bound. The second is the Wasserstein distance, which is implicitly linked to the Lipschitzness property of a neural network. The third one, which is the major novelty in this paper, it to bridge between a continuous GNN to a discrete graph embedding that has more separation power, such as 1-WL or homomorphism vectors. Combining the three types of techniques yields a meaningful generalization bound. The authors gave a concrete case showing that how improved expressivity benefits/harms generalization, which is quite interesting. They also conduct experiments to support the theory.

**Strengths:**

Overall the theoretical result is sound and help gain better understanding for how expressivity is related to generalization in GNNs. The paper is clearly written, with rigorous math notations and theorems. While not checking the proof, I feel the results are correct and make sense to me. The concrete example is quite interesting and makes this paper potentially useful in practice.

**Weaknesses:**

One major weekness of this paper is that, similar to most of the previous generalization papers, the bound given by this paper is quite sophisticated, intractable to compute, and potentially lose. It may be hard to draw the conclusion that this **upper bound** indeed reflects the real case. Another weakness is the readability. The authors did not explain the insights behind this bound clearly. I think it's not intuitive to understand why generalization error can be bounded by the discrete classifier with better separation power. What does the better separation power play a role in the bound? Also, why do the authors rely on Wasserstein distance? I should acknowledge that I am not an expert in generalization theory, but it would be nice to make the paper friendly to the expressivity community as well.

**Questions:**

See the comments above.

---

> ### Author Response · Authors · 2024-11-24
> **Thanks**
>
> > S1 Overall the theoretical result is sound and help gain better understanding for how expressivity is related to generalization in GNNs. The paper is clearly written, with rigorous math notations and theorems. While not checking the proof, I feel the results are correct and make sense to me. The concrete example is quite interesting and makes this paper potentially useful in practice.
>
> Thank you for the feedback. We are glad you liked our presentation and found the results and example interesting.

---

> ### Author Response · Authors · 2024-11-24
> **Computability of bound**
>
> > W1 One major weakness of this paper is that, similar to most of the previous generalization papers, the bound given by this paper is quite sophisticated, intractable to compute,
>
> Thank you for the comment. We want to clarify that alongside the theoretical bound on the generalization gap in Theorem 5.1, we also provide a sample-based version of the bound which is computable. This bound can be found in Appendix D and it is the bound we use in our experiments. We have made this more clear in the paper (after Theorem 5.1) by saying
> “More specifically, we show in Appendix D how an efficient to compute sample-based bound can be used instead of the theoretical bound presented in Theorem 5.1. Importantly, we use this practical bound in our experiments.”
> To make matters clearer, in the new Appendix E, we explain how the bounds in our experiments are computed, including VC based bounds, PAC-Bayesian-based bounds and our bounds. In short, we compute our empirical bounds based on Thm D.2, where we use samples from the training set to estimate $\text{Lip}(f)$ and $\Delta(\lambda_\sharp(\mu_c))$. The most computational expensive step is to compute the Wasserstein distance, which has complexity $\mathcal O((\frac{m_c}{2n})^3)$ but is tractable in practice. We have added pseudocode to compute the bound in Algorithm 1 in Appendix E.

---

> ### Author Response · Authors · 2024-11-24
> **Tightness of bounds**
>
> > W2 The bound is potentially loose.
>
> Compared with the VC bound and PAC-Bayesian bound (newly added in revision in response to reviewer Reviewer 31fD), our bound does not grow with layers and thus is more useful in practice. Also, our bound is able to reflect the influence of normalization which cannot be captured by the other two bounds.
> The tightness of our bound varies on datasets. There are two major dataset-dependent factors in Theorem D.2 that influence the sharpness of the bound: $m_c$ (samples in each class), and $K$ (number of classes). ENZYMES has 6 classes but only has 600 graphs, leading to a small $m_c$ that increases the bound. In contrast, SIDER (1427 graphs 2 classes),  resulting in a smaller and tighter bound. Because of this, to evaluate the bound, it would make more sense to observe how the bound reflects loss gaps across different layers and choices of homomorphism patterns. This also aligns well with real-world scenarios where a dataset is often given and a practitioner is interested in the influence of parameters and model choices on generalization. In the newly added correlation matrices in Table 2, we note our bound correlates well with empirical gaps.

---

> ### Author Response · Authors · 2024-11-24
> **Alignment with reality**
>
> >W3 It may be hard to draw the conclusion that this upper bound indeed reflects the real case.
>
> In the newly added correlation matrices in Table 2, it can be seen our bound correlates well with empirical gaps. We have also added a new plot (Figure 2) in Appendix F which shows how our bounds change with loss gaps across layers. We also note in Figure 1 that our bound can well reflect the influence of homomorphism patterns on loss gaps. This highlights the potential of using our bound to guide model design and hyperparameter selection.
>
> That said, the concern raised by the reviewer is a general concern related to the theory and practice of generalization. It is, undoubtedly, one of the main open questions in deep learning (and graph learning) to find complexity measures that accurately predict generalization error. In fact, one of the outcomes of the NeurIPS 2020 Competition: Predicting Generalization in Deep Learning was that there is a need for more rigorous theoretical bounds [1]. The work by Chuang et al. [2], on which our bound is based, shows that they outperform other bounds in most (six out of eight) benchmark tasks from the Competition mentioned earlier. Although those tasks are related to general deep learning, their k-Variance bound is one of the best ones for predicting generalization errors [2]. This somewhat justifies why we base our work on the k-Variance-based bounds.
>
> Of course, more research is needed to align theory and practice even better, both for graph learning and for general deep learning. It is indeed hard to draw the conclusion that  theoretical bounds reflect the real world. However, as mentioned already, we do observe experimentally that it aligns well  with the real generalization error (see experimental section, tables ...)
>
> [1] Jiang et al., NeurIPS 2020 Competition:Predicting Generalization in Deep Learning
>
> [2] Chuang et al., Measuring Generalization with Optimal Transport. NeurIPS 2021

---

> ### Author Response · Authors · 2024-11-24
> **Intuition**
>
> > W4. is the readability. The authors did not explain the insights behind this bound clearly. I think it's not intuitive to understand why generalization error can be bounded by the discrete classifier with better separation power. What does the better separation power play a role in the bound? Also, why do the authors rely on Wasserstein distance?
>
> We thank the reviewer for the comments, which we next address in turn.
>
> *Impact of separation power.*  We show that the generalization of GNNs is influenced by  a discrete graph encoder that bounds the expressivity of the GNN. The influence, facilitated by inter-class separation and intra-class concentration of the discrete graph embeddings (Section 5), is possible because of a connection between the embeddings of the encoder and the GNN (Lemma 4.2, Prop 4.3, Cor 4.5). Specifically, the Wasserstein distance of the GNN embeddings is bounded by the embedding of the more powerful graph encoder by a factor $B/S$ (or $\text{Lip}(f)$). This result shows that, intuitively, if the encoder separates graphs in a way that aligns well with class distribution, i.e. concentrated within each class and well-separated between classes (measured by Wasserstein distance), the GNN can potentially generalize well.
>
> In other words, the generalization ability of a GNN can be analyzed by looking at the graph encoder (1-WL, k-WL, F-WL, etc) that bounds it. Such discrete graph encoders are often fast to run and easy to analyze. Hence this insight can be used to guide model design and hyperparameter selection without actually training the model.
>
> *Choice of Wasserstein distance.*  For the choice of Wasserstein distance, we follow the work by Chuang et al. [1], where a generalization bound is obtained in terms of k-variance [2], which in turn is defined in terms of the Wasserstein distance. We  adopted that distance as well. One of the key motivations for using it is that it is known to capture the variance well for clustered distributions. In our setting, the distributions are over the learned features, which are - hopefully - clustered based on their label. Intuitively, it is used to better capture the structural properties of the feature distributions, as mentioned in Chuang et al. There are, of course, also mathematical reasons. For example, the choice of Wasserstein distance in k-variance implies that k-variance shares many properties with the standard notion of variance (see [2] Proposition 2 for details). A final motivation is that the Wasserstein distance based bound was shown by Chuang et al. to predict generalization well [1].
>
>
> [1] Chuang et al., Measuring Generalization with Optimal Transport. NeurIPS 2021
>
> [2] Solomon et al, k-Variance: A Clustered Notion of Variance. SIAM Journal on Mathematics of Data Science, 2022.

---

> > ### Comment · Reviewer_yEuE · 2024-11-25
> > **Thank you**
> >
> > Thank you for the very detailed response! I am generally satisfied with the response and am happy to see the paper accepted. That is to say, I am not an expert in generalization and is not familiar with the key related works and concepts, so I will not change the confidence level.

---

> ### Author Response · Authors · 2024-11-25
>
> Thanks for reading our responses and your support.

---

### Official Review · Reviewer_iUbN · 2024-11-03

**Soundness:** 3
**Presentation:** 4
**Contribution:** 3
**Rating:** 8
**Confidence:** 3

**Summary:**

This work present a practical framework for analyzing the generalization property of GNNs via their expressivity. The main result is an extension of previous works to graph representation learning. The theoretical results are accompanied by case studies and experimental verifications to demonstrate the tightness of the proposed bound.

**Strengths:**

1. The proposed framework is empirically effective to estimate the generalization of GNNs. The bound is given by the regularity and expressivity of the graph encoder itself, which are accessible measures in practice. This method has the potential to assess the generalization of real-world models.

2. The presentation of the paper is clean and easy to follow.

**Weaknesses:**

1. The verification and discussion are limited to MPNNs. Empirically, higher-order information and/or attention mechanism are widely adopted to enhance GNNs. While the results in this work are applicable to these models in a straightforward way, it does not provide insights how these architectures improve the B/S constants in the bound.

**Questions:**

Is it possible to do a priori analysis on how a specific message aggregation scheme, e.g., k-WL or attention, can quantitively influence the terms in the bound?

---

> ### Author Response · Authors · 2024-11-24
> **Impact of expressivity**
>
> We thank the reviewer for the very positive assessment of the paper.
>
>
> > W. The verification and discussion are limited to MPNNs. Empirically, higher-order information and/or attention mechanism are widely adopted to enhance GNNs. While the results in this work are applicable to these models in a straightforward way, it does not provide insights how these architectures improve the B/S constants in the bound.  Is it possible to do a priori analysis on how a specific message aggregation scheme, e.g., k-WL or attention, can quantitatively influence the terms in the bound?
>
> We thank the reviewer for this interesting question, which is a question we considered while writing the paper. In general, it is difficult to draw conclusions since the influence of architectural design of the graph model  is dependent on the graph data distribution. However, with some minor assumptions we can provide some insights, as follows.
>
> We first explain why the influence on generalization is data-dependent. While theoretically the k-WL is 1-separating considering all graphs, for a given data distribution, it can have a greater separation than 1-WL and hence a larger $S$ which makes $B/S$ potentially smaller. However, the Wasserstein distance term in the bound for k-WL is not smaller than 1-WL. This is because k-WL can separate all graphs 1-WL can separate and increase the differences in color histogram, e.g. if the color histograms of G and H differs by 1 in 1-WL (only one vertex has a different color), there are at least $\min(|V_G|, |V_H|)^{k-1}$ vertex tuples would have different colors in k-WL. So it boils down to the trade off between the decrease in B/S and the increase in intra-class Wasserstein distance.
>
> Suppose now that we fix $B$ and assume that we map graphs into $[0,B]$. That is, we assume all embeddings to be $B$-bounded (e.g., using normalized versions of k-WL, k-MPNNs etc). Suppose that we work in a discrete setting and that there are $m_k$ distinguishable graphs in G using k-WL. Then, maximal separability is $B/m_k$ when all $m_k$ are spread evenly over $[0,B]$. This results in a factor $(B/(B/m_k))=m_k$. Since $m_k$ increases with $k$, a smaller $k$ is preferred. Also a small $k$ also keeps the intra-class Wasserstein distance term small so it is advantageous to choose the smallest $k$.
>
> In terms of attention GNNs, for Graph Attention Networks (GAT) which uses attention in the feature update function and fall into the category of MPNNs. So our results regarding MPNNs apply to it. The second type is graph transformers which adopts the transformer architecture and uses structural encodings to capture graph structure information. Graph transformers’ ability to distinguish graphs largely depends on the structure encodings they use (subtree extractor, subgraph extractor, WL trees, etc.). Some of these encodings have known expressivity bounds (e.g. 1-WL) [1]. For these our framework can be used to evaluate their generalization similarly to our analysis in the paper.
>
> [1] Chen et al., Structure-Aware Transformer for Graph Representation Learning. ICML 2022.

---

### Author Response · Authors · 2024-11-24
**Thank you**

We sincerely thank the reviewers for taking the time to read our submission and for providing positive feedback. If there are any additional questions or clarifications needed, please do not hesitate to let us know. Should our responses adequately address your concerns, we would greatly appreciate your consideration of a higher score. Thank you once again for your thoughtful reviews.

---

### Meta-Review · Area_Chair_XdA7 · 2024-12-23

**Metareview:**

This paper studies expressivity and generalization of graph neural networks (GNNs). The generalization bound is characterized by the ratio between the concentration of intra-class embeddings and the separation of inter-class embeddings measured by the Wasserstein distance. The theoretical analysis is supported by some numerical experiments.

The analysis is novel and provides a non-trivial insight. The analysis successfully characterizes the relation between expressivity and generalization with solid and convincing mathematical reasoning. This is a new perspective in the literature.
Moreover, the empirical experiments well justifies the theoretical analysis, for example, the derived bounds are well aligned with the actual loss more than the VC bounds.

For these reasons, this paper can be accepted by ICLR.

**Additional Comments On Reviewer Discussion:**

The reviewers pointed out some concerns about readability and missing derivation of some equations. However, those concerns were properly addressed by the authors. The paper is overall supported by the reviewers.

---

### Decision · Program_Chairs · 2025-01-22

Accept (Poster)